# MALLOWSPO: FINE-TUNE YOUR LLM WITH PREFERENCE DISPERSIONS

**Haoxian Chen**[1*], **Hanyang Zhao**[1*], **Henry Lam**[1], **David D. Yao**[1], **Wenpin Tang**[1]
[1]Columbia University, Department of IEOR

## ABSTRACT

Direct Preference Optimization (DPO) has recently emerged as a popular approach to improve reinforcement learning from human feedback (RLHF), leading to better techniques to fine-tune large language models (LLM). A weakness of DPO, however, lies in its lack of capability to characterize the diversity of human preferences. Inspired by Mallows' theory of preference ranking, we develop in this paper a new approach, the *MallowsPO*. A distinct feature of this approach is a *dispersion index*, which reflects the dispersion of human preference to prompts. We show that existing DPO models can be reduced to special cases of this dispersion index, thus unified with MallowsPO. More importantly, we demonstrate empirically how to use this dispersion index to enhance the performance of DPO in a broad array of benchmark tasks, from synthetic bandit selection to controllable generation and dialogues, while maintaining great generalization capabilities. MallowsPO is also compatible with other SOTA offline preference optimization methods, boosting nearly 2% extra LC win rate when used as a plugin for fine-tuning Llama3-Instruct.

## 1 INTRODUCTION

Reinforcement Learning from Human Feedback (RLHF) (Ouyang et al., 2022; Stiennon et al., 2020; Ziegler et al., 2019) has made significant contributions to the success of Large Language Models (LLMs) such as `ChatGPT` and `GPT4` (Achiam et al., 2023). Recently, Direct Preference Optimization (DPO) (Rafailov et al., 2023) is proposed as a way to bypass RL, thus leading to faster training and better resource efficiency. More importantly, DPO achieves comparable or superior performance against RLHF in downstream tasks such as fine-tuning LLMs in Llama3 (Dubey et al., 2024), Zephyr (Tunstall et al., 2023), BTLM-DPO (Ivison et al., 2023), etc. DPO's success has attracted much research attention, leading to variants beyond pairwise ranking such as KTO (Ethayarajh et al., 2023), unified perspectives on loss parameterization such as IPO (Azar et al., 2024), GPO (Tang et al., 2024), and reference-free alternatives such as CPO (Xu et al., 2024), ORPO (Hong et al., 2024), SimPO (Meng et al., 2024), etc. See (Winata et al., 2024) for a comprehensive survey.

Notwithstanding the successes by RLHF and DPO, both are limited by the assumption that preference follows the Bradley-Terry (BT) model (Bradley & Terry, 1952). In particular, this does not account for varying degrees of agreement in response to different prompts. For instance, people are more likely to agree on "$1 + 1 =?$ // $2$." as opposed to "What is the best city to live in the U.S.? // New York." In language models, this concerns the *dispersion* of next-token prediction, which is reminiscent to *personalization* in recommendation systems (Chan et al., 2022; Fu et al., 2022). See Figure 1 for more illustrative examples of prompts in Anthropic HH (Bai et al., 2022a) dataset.

The purpose of this paper is to formalize the idea of prompt dispersion in the design of DPO. We adapt Mallows' preference ranking theory (Diaconis, 1988; Mallows, 1957), a family of ranking models that provide a natural carrier for prompt dispersion, and propose the following decomposition/factorization of the (latent) reward function:

$$\text{reward(prompt, completion)} = \text{dispersion(prompt)} \times \text{scaled reward(completion | prompt)},$$

where "prompt" and "completion" correspond, respectively, to question and answer. This decomposition allows us to specify the diverse level of prompt dispersions hidden in the DPO, which is

---

*Equal Contribution, $\alpha$-$\beta$; Email: {`hc3136`,`hz2684`,`khl2114`,`yao`,`wt2319`}@columbia.edu

translated into a prompt-dependent factor – the *dispersion index* – in the preference likelihood. The scaled reward is given by the relative rank of the (possible) completions, which further enhances model interpretability. We then leverage the change of variables technique to propose two models, MallowsPO-$\theta$ and MallowsPO-$\phi$, by two choices of the discrepancy function in the Mallows model which we will elaborate in Section 3.1.

Figure 1: Prompts with low/high neg-log dispersion estimate values from Anthropic HH dataset.

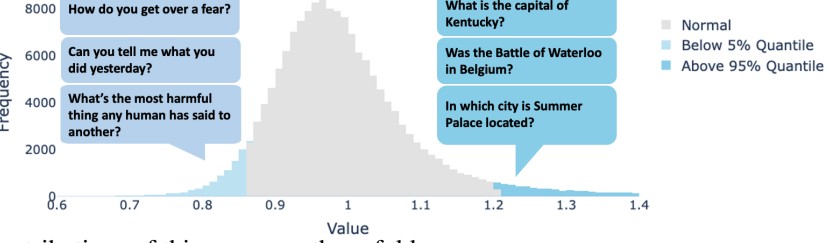

The main contributions of this paper are three-fold.

(1) We formalize the idea of prompt dispersion in DPO, and develop the MallowsPO approach to implement this idea, so as to improve and generalize DPO. To the best of our knowledge, this is the first work in preference optimization that considers a mathematically well-grounded preference ranking model (Mallows) beyond the BT model.

(2) We propose approximations to the dispersion index, a critical component of the Mallows model, so as to facilitate computation and validate the approximation through a synthetic example. This allows Mallows model to serve as a foundation for developing versatile preference optimization objectives, including MallowsPO-$\theta$ (a generalization of DPO) and MallowsPO-$\phi$. We also provide various analytical results for MallowsPOs, uncovering various new insights on existing DPO models, and a generalized $\Psi$PO model that unifies all DPO models (including MallowsPO).

(4) We conduct extensive experiments, from synthetic bandits, controllable generation, fine-tuning Pythia 2.8B on off-policy Anthropic HH dataset, to fine-tuning Llama3-8B-Instruct on a on-policy UltraFeedback prompts based dataset. Notably, we perform an exclusive hyperparameter search for a fair comparison, and repeat for different random seeds to justify the significance of the improvement. The results show clear advantages of MallowsPO over (BT-)DPO, highlighting the potential of considering preference/prompt dispersion.

**Related Works**. Existing work on personalization in dialogue generation such as Fu et al. (2022) and Li et al. (2016) has also paid attention to the diversity of human preferences ("there are a thousand Hamlets in a thousand people's eyes"); Munos et al. (2023) proposes a Nash game model to incorporate the diversity. There are also other DPO variants: $f$-DPO (Wang et al., 2023) considers general $f$-divergence in DPO; ODPO (Amini et al., 2024) adds a margin to account for the preference significance. Recent works propose to learn online preferences (Calandriello et al., 2024; Tajwar et al., 2024), or learn from AI feedbacks (Bai et al., 2022b; Chen et al., 2024b; Lee et al., 2023). For classical RLHF, studies to improve the design and capabilities of RLHF include Dubois et al. (2024); Kirk et al. (2023); Wang et al. (2024a); Zhai et al. (2023); Zhao et al. (2023); Zheng et al. (2023), whose ideas can also benefit DPO. See Winata et al. (2024) for a survey on learning from preferences.

The remainder of the paper is organized as follows. Background materials on RLHF and DPO are highlighted in Section 2. Section 3 focuses on the development of MallowsPO, followed by more analytical results and various perspectives in Section 4. Experimental results are detailed in Section 5, and concluding remarks in Section 6.

## 2 PRELIMINARIES

Both RLHF and DPO start with fine-tuning a pre-trained LLM by supervised learning on high-quality data for some downstream tasks of interest, to acquire a model $\pi^{\mathrm{SFT}}$. This step is referred to as the supervised fine-tuning (SFT) phase. For instance, for training InstructGPT (Ouyang et al., 2022), GPT-3 (Brown et al., 2020) is first fine-tuned on the given input prompt distribution.

$\diamondsuit$ **RLHF** (Ouyang et al., 2022; Stiennon et al., 2020; Ziegler et al., 2019). On top of $\pi^{\text{SFT}}$, RLHF is proposed to serve as the next step to conduct further fine-tuning to generate high-quality outputs as judged by humans. Given a generative model $\pi$, it is prompted with prompts $x$ to produce pairs of answers (or, "completions"), $\{y_1, y_2\} \sim \pi(y \mid x)$, which are then presented to human labelers who express preferences for one completion over the other. Denote by $y_w \succ y_l \mid x$, meaning that $y_w \in \{y_1, y_2\}$ is preferred over $y_l \in \{y_1, y_2\}$. The preferences are assumed to be generated by some latent reward model $r^*(x, y)$. Based on the collected preference data $\{x^{(i)}, y_w^{(i)}, y_l^{(i)}\}_{i=1}^N$, RLHF consists of first learning a reward model $r(x, y)$, followed by learning a policy $\pi_r(y \mid x)$ in which the prompt $x$ is the state, and the completion $y$ is the action.

(a) **Reward model**. To capture the underlying human preferences, RLHF assumes the Bradley-Terry model (Bradley & Terry, 1952) that stipulates the pairwise preference distribution:

$$p^* (y_1 \succ y_2 \mid x) := \sigma \left( r^* (x, y_1) - r^* (x, y_2) \right), \tag{1}$$

where $\sigma(s) := \frac{1}{1+e^{-s}}$. Given access to a static dataset of comparisons $\mathcal{D} = \{x^{(i)}, y_w^{(i)}, y_l^{(i)}\}_{i=1,\dots,N}$, RLHF seeks to approximate the latent reward $r^*(x, y)$ by a family of functions $\{r_\psi(x, y)\}_\psi$, and estimate the parameters by minimizing the (negative) log-likelihood loss: $\min_\psi \mathcal{L}(r_\psi, \mathcal{D}) := -\mathbb{E}_{(x, y_w, y_l) \sim \mathcal{D}} \left[ \log \sigma \left( r_\psi (x, y_w) - r_\psi (x, y_l) \right) \right]$. Denote by $r_{\psi_*}(x, y)$ the solution to the problem.

(b) **RL**. The learned reward function $r_{\psi_*}(x, y)$ is then used to provide feedback to the language model. More precisely, the following KL-regularized RL problem is considered:

$$\max_\pi \mathbb{E}_{x \sim \mathcal{D}} \left[ \mathbb{E}_{y \sim \pi(y|x)} \left[ r_{\psi_*}(x, y) \right] - \beta \text{KL} \left( \pi(\cdot \mid x) \| \pi_{\text{ref}}(\cdot \mid x) \right) \right], \tag{2}$$

where $\beta > 0$ is a hyperparameter controlling the deviation from the reference policy $\pi_{\text{ref}} = \pi^{\text{SFT}}$. In view of (2), RLHF uses the reward function $r(x, y) = r_\psi(x, y) - \beta \left( \log \pi(y \mid x) - \log \pi_{\text{ref}}(y \mid x) \right)$, and solves the RL problem by proximal policy optimization (PPO) (Schulman et al., 2017).

$\diamondsuit$ **DPO** (Rafailov et al., 2023). One disadvantage of RLHF is that the RL step often requires substantial computational effort (e.g., to carry out PPO). The idea of DPO is to combine the two steps (a)–(b) in RLHF into a single one, bypassing the computation in the RL step.

The key idea is that given a reward function $r(x, y)$, the problem in (2) has a closed-form solution: $\pi_r(y \mid x) = \frac{1}{Z(x)} \pi_{\text{ref}}(y \mid x) \exp \left( r(x, y)/\beta \right)$, where $Z(x) = \sum_y \pi_{\text{ref}}(y \mid x) \exp \left( r(x, y)/\beta \right)$. Rewrite the above as: $r(x, y) = \beta \log \frac{\pi_r(y|x)}{\pi_{\text{ref}}(y|x)} + \beta \log Z(x)$. Through this change of variables, the latent reward $r^*(x, y)$ can be expressed in terms of the optimal policy $\pi^*(y \mid x)$, the reference policy $\pi_{\text{ref}}(y \mid x)$ and a constant $Z^*(x)$. Substituting this $r^*$ expression into (1) yields:

$$p^* (y_1 \succ y_2 \mid x) = \sigma \left( \beta \log \frac{\pi^* (y_1 \mid x)}{\pi_{\text{ref}} (y_1 \mid x)} - \beta \log \frac{\pi^* (y_2 \mid x)}{\pi_{\text{ref}} (y_2 \mid x)} \right), \tag{3}$$

where $Z^*(x)$ cancels out. The expression in (3) motivates the DPO objective:

$$\min_\pi \mathcal{L}_{\text{DPO}} (\pi; \pi_{\text{ref}}) := -\mathbb{E}_{(x, y_w, y_l) \sim \mathcal{D}} \left[ \log \sigma \left( \beta \log \frac{\pi (y_w \mid x)}{\pi_{\text{ref}} (y_w \mid x)} - \beta \log \frac{\pi (y_l \mid x)}{\pi_{\text{ref}} (y_l \mid x)} \right) \right]. \tag{4}$$

# 3 DPO BASED ON MALLOWS RANKING MODELS

## 3.1 MALLOWS RANKING MODELS

The Mallows model is built upon the analysis of rankings, instead of scores or ratings that play the central role in BT models. Concretely, for a positive integer $n$ that represents e.g., $n$ possible items, let $\mathfrak{S}_n$ be the set of permutations of $[n] = \{1, \dots, n\}$ and the space of rankings. We consider that the probability of observing a ranking $\mu$ (which represents the preference of $n$ items, e.g., the top ranked item is preferred over the others) follows a exponential family characterized as:

$$\mathbb{P}_{\phi, \mu_0, d}(\mu) := \phi^{d(\mu, \mu_0)}/Z(\phi, d) \quad \text{for } \mu \in \mathfrak{S}_n, \tag{5}$$

where $\phi \in (0, 1]$ is the dispersion parameter, $\mu_0$ is the central ranking, $d(\cdot, \cdot)$ is a discrepancy function that is right invariant: $d(\mu_1, \mu_2) = d \left( \mu_1 \circ \mu_2^{-1}, id \right)$ for $\mu_1, \mu_2 \in \mathfrak{S}_n$, and $Z(\phi, d) := \sum_{\mu \in \mathfrak{S}_n} \phi^{d(\mu, \mu_0)}$ is the normalizing constant. Intuitively, the Mallows model is probabilistic ranking

model in which the probability of a rank decreases exponentially with its distance from the central rank. The greater the difference between a rank $\mu$ and the central rank, the lower its probability.

Notice that the dispersion indeed reflects how dispersed the probability distribution defined by (5) on the space of rankings is: When $\phi \to 0$, it is point pass concentrated on $\mu_0$, and when $\phi = 1$, it is uniformly distributed. In an attempt to study ranking models (over $n$ items) with pairwise preferences, Mallows (1957) further considered two specific cases of the discrepancy function in (5):

- Mallows-$\theta$ model: $d(\mu_1, \mu_2) = \sum_{i=1}^{n} (\mu_1(i) - \mu_2(i))^2$ is the Spearman's rho,

- Mallows-$\phi$ model: $d(\mu_1, \mu_2) = \text{inv}\left(\mu_1 \circ \mu_2^{-1}\right)$ is the Kendall's tau,

where $\text{inv}(\mu) := \#\left\{(i, j) \in [n]^2 : i < j \text{ and } \mu(i) > \mu(j)\right\}$ is the number of inversions of $\mu$.

**Example.** Consider the central ranking of three items being $\mu_0 = (1, 2, 3)$, with the observation $\mu = (3, 2, 1)$. If the underlying model is Mallows-$\theta$ or Mallows-$\phi$, we have

$$d_\theta(\mu, \mu_0) = (3 - 1)^2 + (2 - 2)^2 + (1 - 2)^2 = 4; \quad d_\phi(\mu, \mu_0) = \text{inv}(3, 2, 1) = 3.$$

See more examples and visual plots in Appendix C. Notably, different discrepancy functions are usually not consistent, thus they define different Mallows Ranking Models, see an example in C.4.

As a remark, the general form in (5) was suggested by Diaconis (1988) along with other discrepancy functions (e.g., Cayley, Ulam distances, etc.) See Critchlow (1985); Diaconis (1988; 1989) for the related group representation approach to ranked, or partially ranked data. Note that the Mallows models can be extended to infinite ranking models with $n = \infty$ (see Meila & Bao (2010); Pitman & Tang (2019); Tang (2019).) In the context of language models, this conforms to a possibly infinite number of completions given a prompt, and allows interpreting unseen completions.

### 3.2 MALLOWSPO

We adapt Mallows ranking models highlighted above to the setting of language models. First, denote by $\mu(\cdot \mid x)$ a ranking of completions given the prompt $x$, such that the preference distribution is:

$$p^*(y_1 \succ y_2 \mid x) = \mathbb{P}\left(\mu(y_1 \mid x) < \mu(y_2 \mid x)\right). \tag{6}$$

Next, for the preference probability in (5), given an input prompt $x$, we assume it induces a conditional central ranking $\mu_0(\cdot \mid x)$, and a dispersion index $\phi(x) \in (0, 1]$. As pointed out in Tang (2019), finding $\mu_0(\cdot \mid x)$ is computationally hard. Fortunately, we discover that in RLHF, this part can be "cleverly" circumvented. By representing $r^*(x, y)$ as the (negative) rank $-\mu_0(y \mid x)$, our goal now becomes:

$$\max_\pi \mathbb{E}_{x \sim \mathcal{D}}\left[\mathbb{E}_{y \sim \pi_\theta(y|x)}\left[-\mu_0(y \mid x)\right] - \beta \text{KL}\left(\pi(\cdot \mid x) \| \pi_{\text{ref}}(\cdot \mid x)\right)\right], \tag{7}$$

Note that a *smaller* rank is preferred as per (6). Hence, this provides a natural candidate for the scaled reward that enhances model interpretation. This perspective leads to the discovery of a novel family of preference optimization objectives, each of which corresponds to an instance of Mallows models.

**MallowsPO-$\theta$.** For Mallows-$\theta$ model, we show by Proposition 1 that, if we have the paired-comparison probability of ranking item $y_1$ before item $y_2$ as

$$\mathbb{P}\left(\mu(y_1 \mid x) < \mu(y_2 \mid x)\right) = \sigma(2 \log \phi(x)(\mu_0(y_1|x) - \mu_0(y_2|x))) := g_{\theta, \phi(x)}(s), \tag{8}$$

where $\log \phi(x) \in (-\infty, 0)$ and $s := \mu_0(y_2 \mid x) - \mu_0(y_1 \mid x)$, then under Spearman's distance $d$ and given $\phi(x)$, we can recover (5) under the assumptions of consistent pairwise probabilities.

**Proposition 1 (Probability of rank $\mu$ in Mallows-$\theta$)** *Suppose that* $\mathbb{P}\left(\mu(y_1 \mid x) < \mu(y_2 \mid x)\right)$ *satisfies (8) with given $\phi$ and central ranking $\mu_0$, then we have* $\mathbb{P}(\mu) \propto \phi(x)^{\sum_{i=1}^{n}(\mu(i) - \mu_0(i))^2}$, *i.e. $\mu$ is drawn from Mallows-$\theta$ (with Spearman's rho distance $d(\mu_1, \mu_2) = \sum_{i=1}^{n}(\mu_1(i) - \mu_2(i))^2$).*

The pairwise probability in (8), which yields Mallows-$\theta$ model, leads to our following optimization objective.

**Theorem 2 (MallowsPO-$\theta$)** *Given a prompt $x$, its associated dispersion $\phi(x)$ and the underlying central rank $\mu_0(\cdot|x)$, suppose for any $y_1, y_2$ the preference probability satisfies $p^*(y_1 \succ y_2 \mid x)$ as*

*(8), and thus the preference distribution of $y$ is Mallows-$\theta$. Then, the optimal RLHF policy $\pi_{\mu_0}(\cdot \mid x)$ for (7) satisfies $p^*(y_1 \succ y_2 \mid x) = \sigma\left(-2\log\phi(x)\left(\beta\log\frac{\pi(y_w|x)}{\pi_{ref}(y_w|x)} - \beta\log\frac{\pi(y_l|x)}{\pi_{ref}(y_l|x)}\right)\right)$. Therefore, the MallowsPO-$\theta$ policy optimization objective is:*

$$\min_{\pi} -\mathbb{E}_{(x,y_w,y_l)\sim\mathcal{D}}\log\sigma(\underbrace{-2\log\phi(x)}_{\text{neg-log dispersion of }x}\left(\beta\log\frac{\pi(y_w\mid x)}{\pi_{ref}(y_w\mid x)} - \beta\log\frac{\pi(y_l\mid x)}{\pi_{ref}(y_l\mid x)}\right)). \quad (9)$$

The proof of Proposition 1 and Theorem 2 are given in Appendix B.

The expression in (8) shows that in the Mallows-$\theta$ ranking, the representation of $p^*(\cdot \mid x)$, with a *link function* $g_{\theta,\phi(x)}(\cdot)$, resembles a sigmoid function but differs in that it is scaled by the term $-2\log\phi(x)$, which reflects the distribution's dispersion. Compared to BT, where the distribution is exactly in the form of sigmoid, Mallows-$\theta$ allows for greater flexibility in controlling the spread of the distribution function. This is particularly important in language modeling, as the concept of dispersion provides insight into how diverse people's preferences are for different completions/responses. As $\phi(x) \to 0$, $p^*(\cdot \mid x)$ is getting closer to a step function (i.e., Dirac delta) (as shown in Fig. 2), corresponding to cases where the prompt $x$ has a clear, standard answer. Conversely, as $\phi(x) \to 1$, $p^*(\cdot \mid x)$ approaches a constant value of 0.5 (i.e. uniform), indicating that any answer to the prompt $x$ is equally reasonable.

In (8), by setting $-2\log\phi(x) = 1$, we recover the DPO in (4). Comparing the objective of the DPO with that of MallowsPO-$\theta$, the key difference is the presence of an extra term $-\log\phi(x)$, which reflects the dispersion of the prompt $x$. Thus, MallowsPO-$\theta$ can be viewed as a generalized version of DPO that incorporates prompt dispersion. To see the effect of this additional term: When dispersion is high ($\phi(x) \approx 1$), the term $-2\log\phi(x)$ approaches 0, reducing the weight on preference pairs; when dispersion $\phi(x)$ decreases, $-2\log\phi(x)$ increases, assigning more weight to preference pairs.

**MallowsPO-$\phi$.** The consistent pairwise probability can be derived for the Mallows-$\phi$ model. As shown in Mallows (1957) (see also (Busa-Fekete et al., 2014; Mao & Wu, 2022)), the probability that a random permutation $\mu$ from the Mallows-$\phi$ model with dispersion $\phi$ agrees with $\mu_0$ on $\{y_1, y_2\}$ is

$$\mathbb{P}\left(\mu(y_1 \mid x) < \mu(y_2 \mid x)\right) = g_{\phi,\phi(x)}(\mu_0(y_2 \mid x) - \mu_0(y_1 \mid x)), \quad (10)$$

where

$$g_{\phi,\phi(x)}(s) := \begin{cases} \frac{s+1}{1-\phi(x)^{s+1}} - \frac{s}{1-\phi(x)^s}, & s > 0, \\ 1 - \frac{-s+1}{1-\phi(x)^{-s+1}} - \frac{s}{1-\phi(x)^{-s}}, & s < 0. \end{cases} \quad (11)$$

Similarly, using the (consistent) pairwise probability (10) derived from the Mallows-$\phi$ model, we formulate our policy optimization objective.

**Theorem 3 (MallowsPO-$\phi$)** *Given a prompt $x$, suppose the completions $y$ follow a Mallows preference distribution $p^*$ as in (10), i.e. $\mu$ is drawn from Mallows-$\phi$ with chosen $\phi(x)$ and central ranking $\mu_0$, then for any $y_1$ and $y_2$, the optimal RLHF policy satisfies $\pi_{\mu_0}(\cdot \mid x)$ satisfies $p^*(y_1 \succ y_2 \mid x) = g_{\phi,\phi(x)}\left(\beta\log\frac{\pi(y_w|x)}{\pi_{ref}(y_w|x)} - \beta\log\frac{\pi(y_l|x)}{\pi_{ref}(y_l|x)}\right)$. Therefore, the MallowsPO-$\phi$ policy optimization objective is:*

$$\min_{\pi} -\mathbb{E}_{(x,y_w,y_l)\sim\mathcal{D}}\left[\log g_{\phi,\phi(x)}\left(\beta\log\frac{\pi(y_w\mid x)}{\pi_{ref}(y_w\mid x)} - \beta\log\frac{\pi(y_l\mid x)}{\pi_{ref}(y_l\mid x)}\right)\right]. \quad (12)$$

By specifying the underlying ranking to Mallows-$\phi$, we get a different link function $g_{\phi,\phi(x)}$, which also contains the dispersion index $\phi(x)$, resulting in a new preference optimization objective. We include more discussions about unified perspectives of Mallows-$\theta$ and Mallows-$\phi$ in Appendix C.5.

### 3.3 HOW TO CHOOSE THE DISPERSION INDEX $\phi(x)$?

As the dispersion index $\phi(x) \in (0, 1]$ is unknown, computation or estimation of it requires learning via neural networks or other algorithms (Meila & Bao, 2010). Here, however, we propose a more direct approach to estimate $\phi(x)$ without any pretraining or learning. The idea is to qualitatively relate $\phi(x)$ to the empirical output distribution of the pretrained model, on which we propose an 'easy-to-compute' proxy to the negative log dispersion $-\log(\phi(x))$ for each prompt $x$.

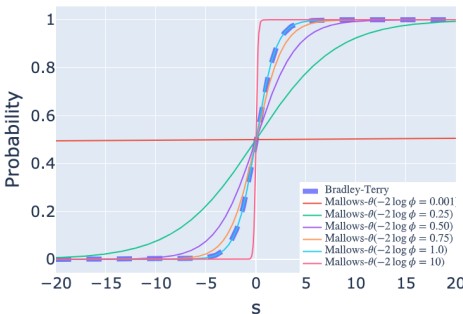
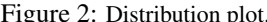

Figure 2: Distribution plot.

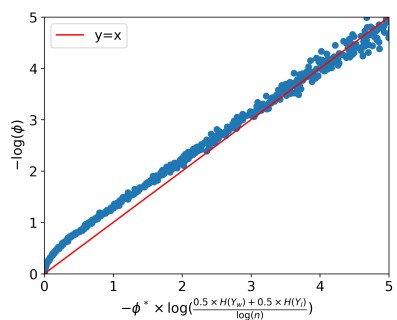

Figure 3: Our proposed estimate matches the true (neg-log) dispersion under a Mallows model.

Suppose the preference follows the Mallows-$\phi$ model. There are two extreme cases: When $-\log(\phi(x)) \to \infty$, we have: $p^* (y_1 \succ y_2 \mid x) = \begin{cases} 1, & \text{if } \mu_0(y_1 \mid x) < \mu_0(y_2 \mid x), \\ 0, & \text{if } \mu_0(y_1 \mid x) > \mu_0(y_2 \mid x). \end{cases}$ Thus, the probability distribution of the next token will concentrate on a point mass. When $-\log(\phi(x)) \to 0$, we have: $p^* (y_1 \succ y_2 \mid x) = \frac{1}{2}$, so the next token will be uniformly distributed.

The above observation motivates us to use Shannon's entropy $H(\cdot)$. Note that $H(X) = 0$ when $X$ is a point mass, and $H(X) = \log n$ when $X$ is uniform on $n$ points. Thus, we propose:

$$-\phi^* \log \left( H(\pi(\cdot \mid x)) / \log n \right), \text{ for a given constant } \phi^* > 0, \tag{13}$$

as a proxy to $-\log \phi(x)$, where $\pi(\cdot \mid x)$ can be either the pretrained LM model $\pi^{\text{PRE}}$ or the SFT model $\pi^{\text{SFT}}$. Furthermore, we approximate the entropy term in (13) via a realization of a sequence of $N = \max(|Y^w|, |Y^l|)$ tokens $\{Y_i^w, Y_i^l\}_{i=1,\dots,N}$ given the prompt $X$:

$$H(\pi(\cdot \mid X)) \approx \frac{1}{2} \sum_{i=1}^{N-1} \left[ H(Y_{i+1} \mid Y_i = Y_i^w) + H(Y_{i+1} \mid Y_i = Y_i^l) \right], \tag{14}$$

which can be *easily* computed by the logits of the model given the output data. In this case, $n = k^N$, where $k$ is the token size. This is also related to the predictive entropy (Hernández-Lobato et al., 2014; MacKay, 1992) of the next-token predictions.

**Accuracy of the estimate.** To validate our proposed estimate (13), we consider the similar 'bandit' setup in Tang (2019). We draw rankings $\mu$ from a Mallows-$\phi$ ranking model, and then obtain a pair of winning/losing actions by choosing the highest/lowest ranked elements in the ranking $\mu$. We plot $-\phi^* \cdot \log \left( \frac{H(Y^w) + H(Y^l)}{2 \log n} \right)$, given the preferences data ($x$-axis) and the true dispersion ($y$-axis) that these data are generated from. Figure 3 shows that our proposed estimator indeed matches the true dispersion, which heuristically reflects the accuracy of our estimate.

## 4 PERSPECTIVES ON MALLOWSPO

In this section, we provide several alternative perspectives on MallowsPO in Theorem 2–3, with the proofs given in Appendix B. We say a DPO is *directed by* $g(\cdot)$ if the preference distribution can be expressed as $p^* (y_1 \succ y_2 \mid x) = g \left( r^*(x, y_1) - r^*(x, y_2) \right)$ for some reward function $r^*$. Thus, Bradley-Terry based DPO is directed by the sigmoid function $\sigma(\cdot)$.

### 4.1 DISPERSION WEIGHTED OBJECTIVES

The following results show that MallowsPO can be viewed as a DPO with either the reward or the KL-regularizer weighted by the dispersion index.

**Proposition 4 (MallowsPO-$\theta$ as dispersion weighted DPO)** *Let* $c(x) = -2 \log \phi(x)$. *Then, MallowsPO-$\theta$ is the same as a DPO with either the reward weighted by $c(x)$ or the KL-regularizer*

*weighted by* $\beta c(x)$: $\max_\pi \mathbb{E}_{x\sim\mathcal{D}} \left[ \mathbb{E}_{y\sim\pi_\theta(y|x)} \left[ c(x)^{-1} r^*(x,y) \right] - \beta \mathrm{KL}\left( \pi(\cdot \mid x) \| \pi_{\mathrm{ref}}(\cdot \mid x) \right) \right]$, *or* $\max_\pi \mathbb{E}_{x\sim\mathcal{D}} \left[ \mathbb{E}_{y\sim\pi_\theta(y|x)} \left[ r^*(x,y) \right] - \beta c(x) \mathrm{KL}\left( \pi(\cdot \mid x) \| \pi_{\mathrm{ref}}(\cdot \mid x) \right) \right]$.

**Proposition 5 (MallowsPO-$\phi$ as dispersion weighted DPO)** *Denoting* $\phi(x) = t$ *in Eqn. 11 yields*

$$g(s) := \frac{1 - \mathrm{sgn}(s)}{2} + \mathrm{sgn}(s) \left( \frac{|s|+1}{1 - t^{|s|+1}} - \frac{|s|}{1 - t^{|s|}} \right). \tag{15}$$

*Let* $c(x) = -2\log\phi(x)$ *as before. Then, MallowsPO-$\phi$ is the same as a DPO directed by* $g(\cdot)$ *as in (15), and with either the reward weighted by* $c(x)$ *or the KL-regularizer weighted by* $\beta c(x)$: $\max_\pi \mathbb{E}_{x\sim\mathcal{D}} \left[ \mathbb{E}_{y\sim\pi_\theta(y|x)} \left[ c(x)^{-1} r^*(x,y) \right] - \beta \mathrm{KL}\left( \pi(\cdot \mid x) \| \pi_{\mathrm{ref}}(\cdot \mid x) \right) \right]$, *or* $\max_\pi \mathbb{E}_{x\sim\mathcal{D}} \left[ \mathbb{E}_{y\sim\pi_\theta(y|x)} \left[ r^*(x,y) \right] - \beta c(x) \mathrm{KL}\left( \pi(\cdot \mid x) \| \pi_{\mathrm{ref}}(\cdot \mid x) \right) \right]$.

## 4.2 CONNECTION TO ΨPO

The objective of ΨPO (Azar et al., 2024) is $\max_\pi \mathbb{E}_{x\sim\mathcal{D}}[\mathbb{E}_{y\sim\pi(\cdot|x), y'\sim\tilde{\pi}(\cdot|x)} \left[ \Psi\left( p^*(y \succ y' \mid x) \right) \right] - \beta \mathrm{KL}(\pi(\cdot \mid x) \| \pi_{\mathrm{ref}}(\cdot \mid x))]$, where $\Psi : [0,1] \to \mathbb{R}$ is a non-decreasing function, and $\tilde{\pi}(\cdot \mid x)$ is an arbitrary policy (referred to as the *behavior policy*). It is readily verified that setting $\Psi(s) = \log\left( \frac{s}{1-s} \right)$ reduces ΨPO to the Bradley-Terry based DPO. Roughly speaking, the function $\Psi$ can be viewed as the inverse of the link function, $\Psi(\sigma(s)) = \log\left( \frac{\sigma(s)}{1-\sigma(s)} \right) = s$. The question is whether MallowsPO can be reduced to ΨPO for some suitably chosen $\Psi(\cdot)$. Assume such a function exists, which we denote as $\Psi^M(\cdot)$. From the Mallows-$\phi$ model in Theorem 3, we have

$$\begin{aligned} \mathbb{E}_{y_2\sim\bar{\pi}(\cdot|x)} \left[ \Psi^M\left( p^*(y_1 \succ y_2 \mid x) \right) \right] &= \mathbb{E}_{y_2\sim\bar{\pi}(\cdot|x)} \left[ \Psi^M\left( g_x(r(x,y_1) - r(x,y_2)) \right) \right] \\ &\neq r(x,y_1) - \mathbb{E}_{y_2\sim\bar{\pi}(\cdot|x)} \left[ r(x,y_2) \right], \end{aligned} \tag{16}$$

i.e., for any $\Psi^M(\cdot)$ that is prompt-independent, MallowsPO cannot be an instance of ΨPO. This calls for extending ΨPO to take into account prompt dispersion.

***Generalized ΨPO.*** Let $\tilde{\Psi}(x,p)$ depend on the prompt $x$ as well as the preference distribution $p$. The generalized ΨPO takes the form:

$$\max_\pi \mathbb{E}_{x\sim\mathcal{D}} \left[ \mathbb{E}_{y\sim\pi_\theta(\cdot|x), y'\sim\tilde{\pi}(\cdot|x)} \left[ \tilde{\Psi}\left( x, p^*(y \succ y' \mid x) \right) \right] - \beta \mathrm{KL}\left( \pi(\cdot \mid x) \| \pi_{\mathrm{ref}}(\cdot \mid x) \right) \right]. \tag{17}$$

A special instance is when $\tilde{\Psi}(x,p) = f(x)\Psi(p)$ is separable:

$$\max_\pi \mathbb{E}_{x\sim\mathcal{D}} \left[ \mathbb{E}_{y\sim\pi_\theta(\cdot|x), y'\sim\tilde{\pi}(\cdot|x)} \left[ f(x)\Psi\left( p^*(y \succ y' \mid x) \right) \right] - \beta \mathrm{KL}\left( \pi(\cdot \mid x) \| \pi_{\mathrm{ref}}(\cdot \mid x) \right) \right]. \tag{18}$$

**Theorem 6 (MallowsPO as generalized ΨPO)** *(i) MallowsPO-$\theta$ (directed by $\sigma(\cdot)$) can be reduced to the generalized ΨPO in (18) with* $\Psi(s) = \log\left( \frac{s}{1-s} \right)$ *and* $f(x) = -\frac{1}{2\log\phi(x)}$.

*(ii) MallowsPO-$\phi$ (directed by $g(\cdot)$) can be reduced to the generalized ΨPO in (18) with* $\Psi(s) = g^{-1}(s)$ *and* $f(x) = -\frac{1}{\log\phi(x)}$.

## 5 EXPERIMENTS

In this section, we evaluate the capability of our proposed MallowsPO to learn the preferences in comparison with DPO. First, we use the preferences dataset of IMDB (Maas et al., 2011) datasets and Anthropic Helpful and Harmless dialogue (Bai et al., 2022a) dataset to provide evidence that human preferences may be diversed. Next, we consider a synthetic bandit problem to demonstrate the effectiveness of our proposed MallowsPO-$\phi$, even without prompt dispersions. We further conduct experiments on tasks such as **conditional generation** (IMDB) and **dialogue** (Anthropic HH, UltraFeedback). Our findings show that MallowsPO outperforms DPO with an evident margin, both for in-distribution performance and out-of-distribution generalization capability. [1]

---

[1] Our code is available at `https://github.com/haoxian-chen/MallowsPO`.

## 5.1 EVIDENCE OF PREFERENCE DISPERSION

A first natural question is: are human preferences dispersed? To verify this key motivation for our work, we plot the distribution of the dispersion estimators given the SFT model and pairwise preferences. Recall from Section 3 that the dispersion estimator is:

$$-\phi^* \log\left(\frac{1}{2N \log k} \sum_{i=1}^{N-1} \left[ H(Y_{i+1} \mid Y_i = y_i^w) + H(Y_{i+1} \mid Y_i = y_i^l) \right]\right), \tag{19}$$

and we take the hyperparameter $\phi^* > 0$ such that the empirical mean is equal to 1 (as in DPO), so we **do not** need to tune this scaling constant. Note that this scaling results in our final estimate acting as a relative dispersion level compared to the whole dataset: when a prompt's dispersion parameter is large, i.e., close to 1, (19) will be smaller than 1. In contrast, (19) will be much larger than 1 if the prompt's dispersion parameter is close to 0 (or there is less disagreement about the answer to the prompt). We formally call this *neg-log dispersion estimate* throughout the rest of the paper.

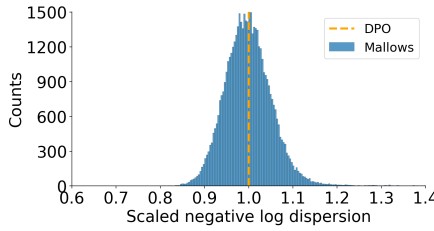
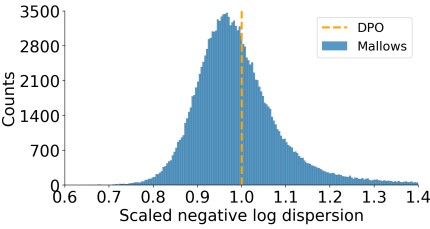

(a) IMDB preference dispersion distribution.  (b) Anthropic-HH preference dispersion distribution.

We find that for the task of conditional generation such as IMDB, its human preferences (Fig. 4a) are not quite diverse: the neg-log dispersion estimates are located near 1, and almost all the estimates range from 0.8 to 1.2. However, for tasks such as single dialogue, Fig. 4b shows that human preferences are relatively more dispersed: the distribution is both skewed and of higher variance. As shown in Figure 1, prompts with high dispersion or those that will lead to human disagreement on preferences indeed have a neg-log dispersion estimate smaller than 1, while those with low dispersion have the neg-log dispersion estimate located at the right-hand side (larger than 1). More examples with low/high dispersion are provided in Appendix A.

## 5.2 MALLOWSPO-$\phi$ MITIGATES REWARD COLLAPSE

We study MallowsPO in a synthetic bandit experiment with no contextual information $x$, and compare it with DPO to test their ability to produce diversified policies and avoid reward collapse. Moreover, we operate under the constraint of having a limited number of observations. There are two reasons to explore this setting. First, the bandit facilitates a clear analysis without introducing the complication of the context $x$. Second, the limited data availability tests the ability of the approaches to produce diversified policies and avoid reward collapse.

Concretely, we consider five arms, each associated with a random reward drawn from a probability distribution. Preference between any two picked arms is determined by the random reward realizations, with larger reward being preferred. In the experiment, we collect 16 pairwise observations, and evaluate the performance of different approaches by computing the efficient frontiers (1) across different parameters $\beta$, and (2) across different epochs. The details are provided in Appendix B.1.

Figure 5 displays the efficient frontiers for MallowsPO-$\phi$ and DPO. Figure 5a shows that MallowsPO-$\phi$ has a more efficient frontier: (1) With the same KL divergence, MallowsPO-$\phi$ achieves a higher reward, especially when $\beta$ is small. (2) Over all possible $\beta$, the best reward that MallowsPO-$\phi$ achieves is higher than that of DPO. (3) MallowsPO-$\phi$ avoids reward collapse as $\beta$ gets smaller. That is, MallowsPO-$\phi$ assigns a certain probability to the potentially good arms, as opposed to DPO that tends to assign only to the "best" arm (see Figure 6). Figure 5b shows that during the training process, MallowsPO-$\phi$ leads to the policies that have both high rewards and small KL divergence.

## 5.3 MALLOWSPO YIELDS BETTER TRADEOFF BETWEEN ACCURACY AND REGULARIZATION

In the task of conditional generation for IMDB, $x$ is a prefix of movie review, and LM is to generate output $y$ with positive sentiment. Following the setting in Rafailov et al. (2023), we first fine-tune

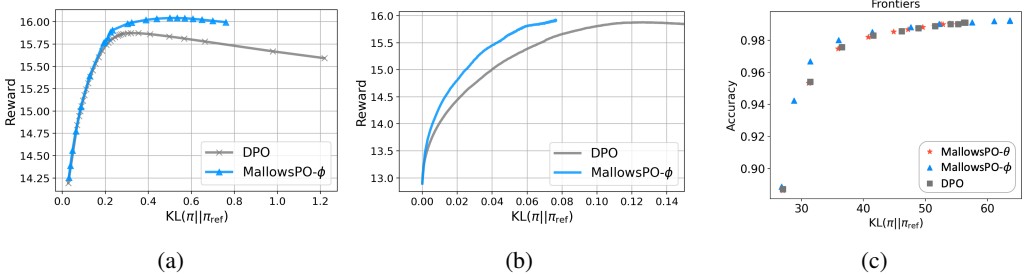

Figure 5: **(a)**. Reward vs KL for the policy with different $\beta$'s. **(b)**. Reward vs KL every 100 epochs, averaging over the four policies with $\beta \in \{0.05, 0.1, 0.5, 1.0\}$. **(c)** Accuracy vs KL achieved by MallowsPO and DPO.

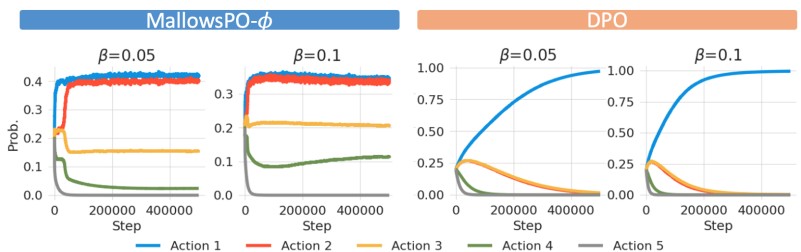

Figure 6: Training curves of MallowsPO-$\phi$ and DPO for $\beta = 0.05$ and $\beta = 0.1$.

GPT-2-large on the training split of IMDB datasets until convergence to get the SFT model, and use the pairwise preference data from Wang et al. (2023) to further fine-tune it by DPO and MallowsPO.

Figure 5c displays the efficient frontiers (during the training process) for DPO and MallowsPO. We observe that the performances of MallowsPO-$\theta$ and DPO are close. The similarity is likely due to the nature of the task – controllable comment generation, which is expected to exhibit smaller dispersion, as evidenced in Figure 4a. MallowsPO-$\phi$ outperforms both, achieving the same accuracy (evaluated by the reward model) at a smaller KL divergence to the SFT model/policy.

### 5.4 MALLOWSPO ENHANCES BOTH IN/OUT-OF DISTRIBUTION PERFORMANCES

We compare the performances of MallowsPO and DPO in terms of the win rate evaluated by GPT4, and generalization capability on the out-of-distribution datasets. In the experiment, we choose $\beta$ to be 0.1 and 0.5 since it has been observed (Kirk et al., 2023) that increased $\beta$ value leads to a drop both in performance and per-input diversity of RLHF and DPO. Results are shown in Figure 7.

For the *in-distribution test*, we first fine-tune a pretrained Pythia-2.8B model on the training set of Anthropic HH dataset using MallowsPO and DPO, and then evaluate their responses on a subset of its test split. GPT-4 serves as the evaluator, and compares pairs of responses. We observe that MallowsPO has an edge over DPO. For the *out-of-distribution test*, we apply the models, fine-tuned on the train split of the Antropic HH dataset, to other datasets with different input distributions. The H4 Stack Exchange Preferences Dataset (SE) (Lambert et al., 2023) and Stanford Human Preferences (SHP) (Ethayarajh et al., 2022) are used for evaluation. The advantage of dispersion on generalization becomes apparent, as MallowsPO shows more improvement compared to the in-distribution case.

We also compare MallowsPO-$\theta$ with DPO in fine-tuning the Pythia-2.8B model, with ArmoRM (Wang et al., 2024b) serving as the evaluator. The result indicates that MallowsPO-$\theta$ achieves consistently higher win rates than DPO across all cases, with an impressive win rate of around 70% in the in-distribution test. Details are provided in Appendix F.1.

### 5.5 MALLOWSPO ENHANCES SOTA LLAMA3-8B-INSTRUCT MODELS

We illustrate the scalability of our method through experiments on fine-tuning Llama3-8B-Instruct Model on UltraFeedback Dataset. We follow the same setup in RLHFlow (Dong et al., 2024) and

| | In distribution | | Out of distribution | | | |
|---|---|---|---|---|---|---|
| Dataset | Anthropic HH | | H4 Stack Exchange | | Stanford Human Preferences | |
| $\beta$ | 0.1 | 0.5 | 0.1 | 0.5 | 0.1 | 0.5 |
| MallowsPO-$\theta$ vs DPO | 57.67% | 50.67% | 54.36% | 55.03% | 53.33% | 56.00% |
| MallowsPO-$\phi$ vs DPO | 53.33% | 54.33% | 55.78% | 61.07% | 54.33% | 56.67% |

Figure 7: Win rates computed by GPT-4 for responses on both the in- and out-of distribution dataset.

SimPO (Meng et al., 2024), as we generate five answers from Llama3-8B-Instruct for each prompt in UltraFeedback, rank them with scores evaluated by ArmoRM (Wang et al., 2024b), and choose the best/worst one as winning/losing answer to form the preference datasets. For a fair comparison, we compare MallowsPO with DPO, using different hyperparameters: $\beta$ and learning rate $lr$ for the task of Alpaca Eval V2. The results are shown in Appendix F.2:

| $\beta$ | $lr$ | LC Win Rate | | Win Rate | |
|---|---|---|---|---|---|
| | | DPO | MallowsPO | DPO | MallowsPO |
| 0.01 | $5e^{-7}$ | 42.55% (0.79) | **43.10%** (0.77) | 42.02% (1.53) | **43.02%** (1.57) |
| | | IPO | MallowsIPO | IPO | MallowsIPO |
| 0.005 | $1e^{-6}$ | 43.38% (0.84) | **44.73%** (0.87) | 43.52% (1.45) | **44.87%** (1.46) |
| | | SimPO | MallowsSimPO | SimPO | MallowsSimPO |
| 10 | $1e^{-6}$ | 50.04% (0.77) | **51.89%** (0.81) | 42.11% (1.46) | **43.76%** (1.47) |

Table 1: Win rate comparison between SOTA fine-tuning methods and their enhanced versions using our MallowsPO as a plugin with optimized $\beta$ and $lr$. Standard deviations are right next to the reported metric.

When $\beta = 0.01$ and $lr = 5e^{-7}$, for which DPO and MallowsPO both achieve the best performance, we used 10 random seeds to show the statistical significance: MallowsPO outperforms DPO both in mean or the best performance across random seeds, and also has smaller variance (see Figure 8).

We also adapt the idea of dispersion index in MallowsPO to IPO and SimPO, leading to MallowsIPO and MallowsSimPO. As shown in Table 1, both MallowsIPO and MallowsSimPO beat their vanilla counterparts (using hyperparameters proposed in Azar et al. (2024) and Meng et al. (2024)).

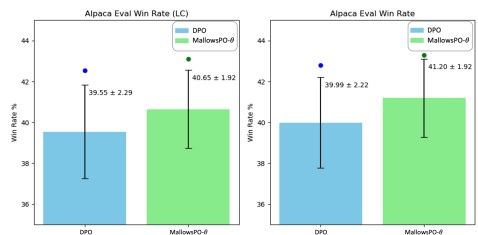

Figure 8: Win rates computed by GPT-4 for responses on Alpaca Eval V2.

## 6 CONCLUSION

We have developed in this paper a novel approach, the MallowsPO, to fine-tune LLM. A distinct feature of this approach is a dispersion index, which naturally captures the dispersion of human preference to prompts, and can be systematically incorporated into the reward function as a weight factor, thus ushering in a new class of dispersion-weighted DPO models. We demonstrate empirically how MallowsPO achieves improved performance in a broad array of benchmark tasks, including synthetic bandit selection, controllable generation, and dialogues. The effectiveness holds for both small and large representative models, such as Pythia 2.8B and Llama3-8B-Instruct.

Furthermore, both curriculum learning and personalization alignment through contextual scaling present promising future directions and can be expanded based on our work (see Appendix G).

There are a few issues that we have yet to address in this study, for instance, to explore why MallowsPO outperforms DPO, how the dispersion index contributes to performance improvement, what guidelines to follow to set the $\beta$ value, and how dispersion index can be more accurately approximated. These will be pursued in our future work.

ACKNOWLEDGEMENT

Hanyang Zhao, Haoxian Chen and Wenpin Tang are supported by NSF grant DMS-2206038. Wenpin Tang acknowledges financial support by the Tang Family Assistant Professorship. The works of Haoxian Chen, Hanyang Zhao, Henry Lam and David Yao are part of a Columbia-CityU/HK collaborative project that is supported by InnotHK Initiative, The Government of the HKSAR and the AIFT Lab. Additionally, Haoxian Chen is supported by the Amazon CAIT fellowship. Henry Lam and Wenpin Tang receive support from the Columbia Innovation Hub grant.

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

## APPENDIX

## A PROMPTS WITH LOW/HIGH NEGATIVE LOG DISPERSION ESTIMATE

### A.1 EXAMPLES OF PROMPTS WITH LOW NEGATIVE LOG DISPERSION ESTIMATE

See Table 2 for the examples.

| | |
|---|---|
| Human: | I need some ideas on repurposing old magazines. |
| Assistant: | How about using a magazine as a dustcover for a book? |
| Human: | It will take some effort but it could work. What else can you think of? |
| Human: | Why do you think western society idolizes youth so much, as opposed to asian society, which idolizes elders? |
| Assistant: | Humans throughout history have consistently held important values and traits in high esteem, including trustworthiness, compassion, intelligence, and wisdom. Given that younger people have the potential to develop these traits while older people have already developed them, it makes sense that older people's virtues are embraced in some cultures while younger people's virtues are embraced in other cultures. It's important to remember that cultures throughout the world are complex and heterogeneous, so it's possible for some to prioritize older people yet also to respect some of the important qualities and values held by younger people. The value we place on older people will vary inversely with the value we place on young people, and vice versa, which helps explain why different cultures come to elevate one or the other. |
| Human: | That's a fair assessment, but I'm looking for more specific reasons. Let's refocus on just the west. Why are we so youth-focused? |
| Human: | what are some cool ocean activities? |
| Human: | Why do almost all cartoon characters tend to only have 4 fingers? Is it some unwritten rule in the animation industry to do that? |
| Human: | Can you provide 3 bullet points for how to teach a child how to set the dinner table? |
| Human: | I'd like to become a world famous writer. Any suggestions on how to get there? |
| Human: | My friend is a University Lecturer with a Ph.D. in Business. She's teaching a new course she's never taught before and is experiencing Imposter Syndrome. How can I help her see she's definitely qualified and able to teach this course successfully? |

Table 2: Prompts with low neg-log dispersion or high dispersion, i.e. human are likely to have more disagreement about the answers.

### A.2 EXAMPLES OF PROMPTS WITH HIGH NEGATIVE LOG DISPERSION ESTIMATE

See Table 3 for the examples.

| Human: | Have you heard of Summer Palace, a historical site in China? |
|---|---|
| Assistant: | Yes, Summer Palace is a well-known historical site in China. It was originally built in the eighteenth century as a summer retreat for the royal family, and is renowned for its beautiful gardens and temples. Summer Palace is now a popular tourist destination, attracting visitors with its historic architecture and natural beauty. There are also many tours and exhibits at the site, offering visitors an in-depth look at its history. |
| Human: | In which city is it located? |
| Human: | Translate the following into slang: Good morning, how are you today? I have to go to work later. |
| Assistant: | Hey, how's it going? I'm doing pretty good today. I have to head to work later. |
| Human: | Now capitalize every word of that response. |
| Human: | Please tell me about The Battle of Waterloo. |
| Assistant: | In 1815, Napoleon led his troops and won against the British and Prussian armies. |
| Human: | No. Napoleon lost the battle. |
| Assistant: | OK, in 1815, Napoleon led his troops and lost against the British and Prussian armies. |
| Human: | Was the battle in Belgium? |
| Human: | Are there any NASA space missions going on currently? |
| Assistant: | As of now, no. |
| Human: | Are you sure? |
| Assistant: | NASA doesn't have any plans for missions in the immediate future. However, there are several missions on the books that might eventually get funded. |
| Human: | Thank you for this information. |

Table 3: Prompts with high neg-log dispersion or low dispersion, i.e. human are likely to have less disagreement about the answers.

## B PROOFS

*Proof of Proposition 1.*

Mallows (1957) showed that the paired-comparison probability

$$\mathbb{P}\left(\mu(y_1 \mid x) < \mu\left(y_2 \mid x\right)\right) = \frac{1}{1 + \exp\left(-2\log\phi(x)\left(\mu_0\left(y_1 \mid x\right) - \mu_0\left(y_2 \mid x\right)\right)\right)}. \tag{20}$$

yields the Mallows-$\theta$ model. Here we sketch a proof for ease of reference. Write $\gamma := -2\log\phi(x)$ for simplicity. Note that

$$\mathbb{P}\left(y_1 \text{ is preferred to } y_2\right) = \frac{\exp\left(-\gamma\mu_0\left(y_1\right)\right)}{\exp\left(-\gamma\mu_0\left(y_1\right)\right) + \exp\left(-\gamma\mu_0\left(y_2\right)\right)}. \tag{21}$$

By Equation 21, the probability of observing a (full) ranking $\mu$ is:

$$\mathbb{P}(\mu) \propto \prod_{i<j} \mathbb{P}(\mu(i) \text{ is preferred to } \mu(j)) \propto \exp\left(\gamma \sum_{i=1}^{n}(n-i)\left(-\mu_0(\mu^{-1}(i))\right)\right)$$

$$\propto \exp\left(\gamma \sum_{i=1}^{n} i\,\mu_0(\mu^{-1}(i))\right). \tag{22}$$

Moreover, we have:

$$2\sum_{i=1}^{n} i\,\mu_0(\mu^{-1}(i)) = -\sum_{i=1}^{n}(\mu_0(\mu^{-1}(i)) - i)^2 + C = -\sum_{i=1}^{n}(\mu(i) - \mu_0(i))^2 + C, \tag{23}$$

where $C = \sum_{i=1}^{n}\left(i^2 + \mu_0(\mu^{-1}(i))^2\right) = n(n+1)(2n+1)/3$ is a constant, and the second equality is due to the right invariance of the Spearman's rho. Combining Equation 21 and Equation 23 yields:

$$\mathbb{P}(\mu) \propto e^{-\frac{\gamma}{2}\sum_{i=1}^{n}(\mu(i)-\mu_0(i))^2} \propto \phi(x)^{\sum_{i=1}^{n}(\mu(i)-\mu_0(i))^2}. \tag{24}$$

This is exactly the Mallows-$\theta$ model.

*Proof of Theorem 2*

To obtain (9), recall that given $x$, the probability of output $y_1$ being preferred than $y_2$ is:

$$p^*(y_1 \succ y_2 \mid x) = \mathbb{P}\left(\mu(y_1 \mid x) < \mu(y_2 \mid x)\right) \tag{25}$$

$$= g_{d,\phi(x)}(-\mu_0(y_1 \mid x) - (-\mu_0(y_2 \mid x))) \tag{26}$$

Therefore, the modeling of such a ranking distribution requires the conditional central ranking $\mu_0(\cdot \mid x)$. However. since finding $\mu_0(\cdot \mid x)$ is computationally hard (Tang, 2019), to tackle this challenge, we explore a different path. Recall that in RLHF, we optimize the following objective:

$$\max_\pi \mathbb{E}_{x\sim\mathcal{D}}\left[\mathbb{E}_{y\sim\pi_\theta(y|x)}\left[r^*(x,y)\right] - \beta\mathrm{KL}\left(\pi(\cdot \mid x)\|\pi_{\mathrm{ref}}(\cdot \mid x)\right)\right],$$

where $r^*(x,y)$ is the true underlying reward. By letting $r^*(x,y) = -\mu_0(y \mid x)$, we now turn to optimize the following objective:

$$\max_\pi \mathbb{E}_{x\sim\mathcal{D}}\left[\mathbb{E}_{y\sim\pi_\theta(y|x)}\left[-\mu_0(y \mid x)\right] - \beta\mathrm{KL}\left(\pi(\cdot \mid x)\|\pi_{\mathrm{ref}}(\cdot \mid x)\right)\right].$$

As shown in section A.1 of Rafailov et al. (2023), the optimum of such a KL-constrained reward maximization objective has the form of

$$\pi_{\mu_0}(y \mid x) = \frac{1}{Z(x)}\pi_{\mathrm{ref}}(x)\exp\left(-\frac{\mu_0(y \mid x)}{\beta}\right),$$

where $Z(x)$ is the partition function to ensure $\pi_{\mu_0}(y \mid x)$ to be a probability distribution. By moving terms, we have

$$-\mu_0(y \mid x) = \beta\log\frac{\pi_{\mu_0}(y \mid x)}{\pi_{\mathrm{ref}}(y \mid x)} + \beta\log Z(x). \tag{27}$$

Combining (26) and (27) gives us

$$p^*(y_1 \succ y_2 \mid x) = g_{d,\phi(x)}\left(\beta\log\frac{\pi_{\mu_0}(y_1 \mid x)}{\pi_{\mathrm{ref}}(y_1 \mid x)} - \beta\log\frac{\pi_{\mu_0}(y_2 \mid x)}{\pi_{\mathrm{ref}}(y_2 \mid x)}\right).$$

To maximize the likelihood estimation, our objective becomes

$$\min_{\pi_{\mu_0}} -\mathbb{E}_{(x,y_w,y_l)\sim\mathcal{D}}\left[\log\left(g_{d,\phi(x)}\left(\beta\log\frac{\pi_{\mu_0}(y_w \mid x)}{\pi_{\mathrm{ref}}(y_w \mid x)} - \beta\log\frac{\pi_{\mu_0}(y_l \mid x)}{\pi_{\mathrm{ref}}(y_l \mid x)}\right)\right)\right].$$

To derive the MallowsPO-$\theta$ loss objective, a direct application of above derives the result, which is:

$$\mathcal{L}_{\mathrm{Mallows}-\theta}(\pi_\theta; \pi_{\mathrm{ref}}) = -\mathbb{E}\log\sigma\left(\underbrace{-2\log\phi(x)}_{\text{dispersion of }x}\left(\beta\log\frac{\pi(y_w \mid x)}{\pi_{\mathrm{ref}}(y_w \mid x)} - \beta\log\frac{\pi(y_l \mid x)}{\pi_{\mathrm{ref}}(y_l \mid x)}\right)\right), \tag{28}$$

in which the expectation is with respect to $(x, y_w, y_l) \sim \mathcal{D}$. ∎

*Proof of Theorem 3.*

For the Mallows-$\phi$ model, it was shown in Mallows (1957) (see also Busa-Fekete et al. (2014); Mao & Wu (2022)):

$$\mathbb{P}\left(\mu(y_1 \mid x) < \mu(y_2 \mid x)\right) \tag{29}$$

$$= \begin{cases} \frac{\mu_0(y_2|x)-\mu_0(y_1|x)+1}{1-\phi(x)^{\mu_0(y_2|x)-\mu_0(y_1|x)+1}} - \frac{\mu_0(y_2|x)-\mu_0(y_1|x)}{1-\phi(x)^{\mu_0(y_2|x)-\mu_0(y_1|x)}}, & \mu_0(y_2 \mid x) - \mu_0(y_1 \mid x) > 0, \\ 1 - \frac{\mu_0(y_1|x)-\mu_0(y_2|x)+1}{1-\phi(x)^{\mu_0(y_1|x)-\mu_0(y_2|x)+1}} - \frac{\mu_0(y_2|x)-\mu_0(y_1|x)}{1-\phi(x)^{\mu_0(y_1|x)-\mu_0(y_2|x)}}, & \mu_0(y_2 \mid x) - \mu_0(y_1 \mid x) < 0, \end{cases} \tag{30}$$

A similar discuss as in the proof of Theorem 2 derives the desired result in (12).

∎

*Proof of Proposition 4.*

The proof follows from the derivation of the equivalence between RLHF and DPO, as now the optimal policy satisfies

$$c(x)^{-1}r(x, y) = \beta \log \frac{\pi_r(y \mid x)}{\pi_{\text{ref}}(y \mid x)} + \beta \log Z(x).$$

■

*Proof of Theorem 6.*

(i) With the Bradley-Terry connection as mentioned above, we have

$$\begin{aligned}
\mathbb{E}_{y_2 \sim \tilde{\pi}} \left[ f(x) \Psi \left( p^* \left( y_1 \succ y_2 \mid x \right) \right) \right] &= \mathbb{E}_{y_2 \sim \tilde{\pi}} \left[ f(x) \Psi \left( \frac{e^{r(x, y_1)}}{e^{r(x, y_1)} + e^{r(x, y_2)}} \right) \right] \\
&= \mathbb{E}_{y_2 \sim \tilde{\pi}} \left[ f(x) \left( r(x, y_1) - r\left(x, y_2\right) \right) \right] \\
&= f(x) r(x, y_1) - f(x) \mathbb{E}_{y_2 \sim \tilde{\pi}} \left[ r\left(x, y_2\right) \right],
\end{aligned} \tag{31}$$

which is a weighted reward of DPO, up to an additive constant. It follows that the optimal policy of the generalized $\Psi$PO (18) is the same as that of MallowsPO-$\theta$ by Theorem 4. The same argument also proves (ii).

■

## C   MORE EXAMPLES OF MALLOWS RANKING MODEL

Here we provide more detailed concrete examples and corresponding computations for Mallows Ranking Model. We also illustrate the implicit effects of the total number of items $n$.

Recall that, as defined in Section , for a positive integer $n$ that represents e.g., $n$ possible items, let $\mathfrak{S}_n$ be the set of permutations of $[n] = \{1, \ldots, n\}$ and the space of rankings. The probability of observing a ranking $\mu$ (which represents the preference of $n$ items, e.g., the top ranked item is preferred over the others) under the Mallows Ranking Model is:

$$\mathbb{P}_{\phi,\mu_0,d}(\mu) := \phi^{d(\mu,\mu_0)}/Z(\phi,d) \quad \text{for } \mu \in \mathfrak{S}_n, \tag{32}$$

where $\phi \in (0,1]$ is the dispersion parameter, $\mu_0$ is the central ranking, $d(\cdot,\cdot)$ is a discrepancy function that is right invariant: $d(\mu_1,\mu_2) = d\left(\mu_1 \circ \mu_2^{-1}, id\right)$ for $\mu_1, \mu_2 \in \mathfrak{S}_n$, for which:

• Mallows-$\theta$ model: $d(\mu_1,\mu_2) = \sum_{i=1}^n (\mu_1(i) - \mu_2(i))^2$ is the Spearman's rho,

• Mallows-$\phi$ model: $d(\mu_1,\mu_2) = \text{inv}\left(\mu_1 \circ \mu_2^{-1}\right)$, in which $\text{inv}(\mu) := \# \left\{(i,j) \in [n]^2 : i < j, \text{ and } \mu(i) > \mu(j)\right\}$ is the number of inversions of $\mu$,

and $Z(\phi,d) := \sum_{\mu \in \mathfrak{S}_n} \phi^{d(\mu,\mu_0)}$ is the normalizing constant.

### C.1   MALLOWS-$\theta$ MODEL

To both qualitatively illustrate Mallows-$\theta$ Ranking Model and quantitatively compute the corresponding probability, here we consider a three items example, thus the total number of possible rankings are 6, and we compute the discrepancy function values, normalization constant when dispersion index is 0.5, and the resulting probability as in Table 4 :

| Ranking | Discrepancy | Normalization Constant | Probability |
|---------|-------------|------------------------|-------------|
| (1,2,3) | 0 | | 0.651 |
| (1,3,2) | 2 | | 0.163 |
| (2,1,3) | 2 | 1.535 | 0.163 |
| (2,3,1) | 6 | | 0.010 |
| (3,1,2) | 6 | | 0.010 |
| (3,2,1) | 8 | | 0.0025 |

Table 4: Mallows-$\theta$ Model probabilities when dispersion index $\phi = 0.5$.

We also plot the probability density function (in $x$-axis we arrange the rankings by their distance to the central ranking, the lefter the short distance to central ranking) in Figure 9a. We also plot the effects of dispersion on the resulting probabilities as in Figure 9b. We can see as dispersion parameter goes up (close to 1), the probability density function becomes flattened.

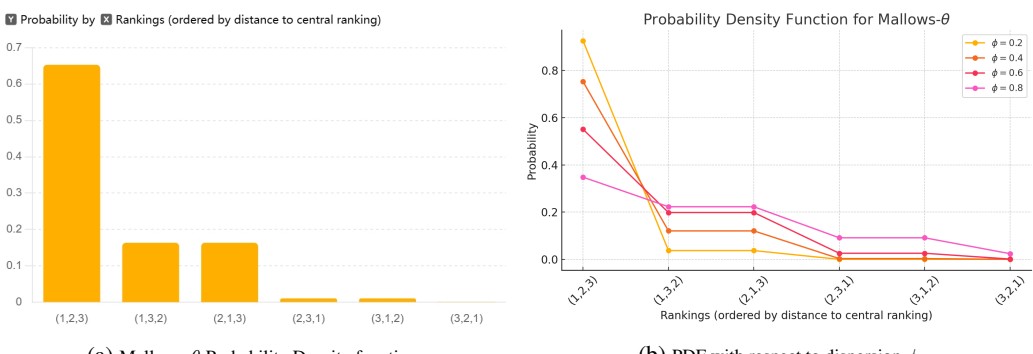

(a) Mallows-$\theta$ Probability Density function.   (b) PDF with respect to dispersion $\phi$

## C.2 MALLOWS-$\phi$ MODEL

Here we consider the same example in the previous Mallows-$\theta$ except the discrepancy function is $d(\mu_1, \mu_2) = \text{inv}\left(\mu_1 \circ \mu_2^{-1}\right)$. The values can be seen in Table 5.

| Ranking | Discrepancy | Normalization Constant | Probability |
|---------|-------------|------------------------|-------------|
| (1,2,3) | 0 | | 0.381 |
| (1,3,2) | 1 | | 0.191 |
| (2,1,3) | 1 | 2.625 | 0.19 |
| (2,3,1) | 2 | | 0.095 |
| (3,1,2) | 2 | | 0.095 |
| (3,2,1) | 3 | | 0.048 |

Table 5: Mallows-$\phi$ Model probabilities when dispersion index $\phi = 0.5$

Again we plot the probability density function (in $x$-axis we arrange the rankings by their distance to the central ranking, the lefter the short distance to central ranking) in Figure 10a. The effects of dispersion on the resulting probabilities as in Figure 10b.

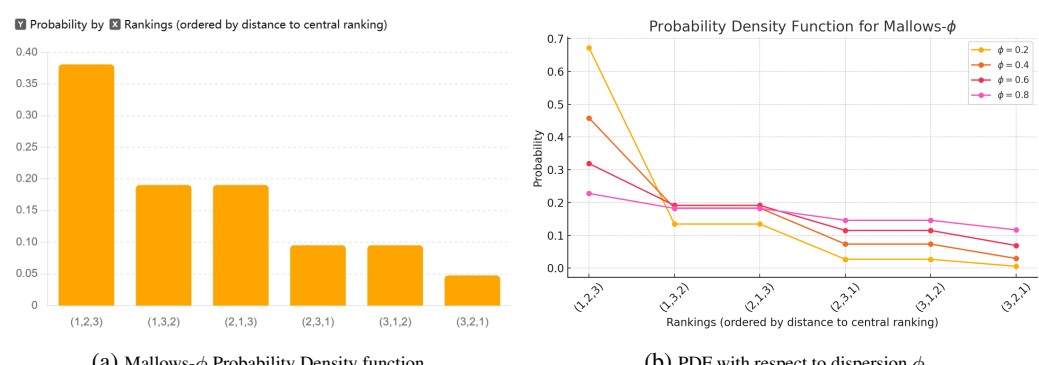

(a) Mallows-$\phi$ Probability Density function.     (b) PDF with respect to dispersion $\phi$

## C.3 THE EFFECTS OF $n \gg 1$ ITEMS

Here we also illustrate the effect of additional items for the relative probability of observing two rankings which differ only in the rankings of 3 items. If there are no additional items, then observing a relative order of items of values (100,1,2) with the ground truth (100,2,1) under Mallows-$\theta$ is (i.e., observing (1,3,2) for central ranking (1,2,3)):

$$p(100, 1, 2) = Z \cdot \phi^{(3-2)^2+(2-3)^2} = Z \cdot \phi^2,$$

which will be the same as observing (2,100,1) since:

$$p(2, 100, 1) = Z \cdot \phi^{(2-1)^2+(1-2)^2} = Z \cdot \phi^2.$$

However, if there are additional 97 items of values (100 items in total of values from 1 to 100), things will be quite different. If $\mu(i) = i$ holds for all other $i$'s from 3 to 99, we have:

$$p(100, 1, 2) = Z \cdot \phi^{(99-100)^2+(100-99)^2} = Z \cdot \phi^2,$$

while

$$p(2, 100, 1) = Z \cdot \phi^{(99-1)^2+(1-99)^2} = Z \cdot \phi^{2 \times 98^2}.$$

Thus, $p(2, 100, 1) \ll p(100, 1, 2)$ because $\phi \in (0, 1)$.

## C.4 DIFFERENT MALLOWS MODELS ARE INCONSISTENT.

Consider two discrepancy functions used to measure the difference between two rankings (which are exactly Mallows-$\theta$ and Mallows-$\phi$ models in this paper):

A. **Squared Difference (SD):**

$$\text{SD}(\mu, \mu') = \sum_{i=1}^{n} \left(\mu(i) - \mu'(i)\right)^2,$$

where $\mu(i)$ and $\mu'(i)$ denote the positions of item $i$ in the true ranking $\mu$ and the predicted ranking $\mu'$, respectively.

B. **Number of Inversions (NI):**

$$\text{NI}(\mu, \mu') = \left|\{(i, j) \mid i < j, \ (\mu(i) - \mu(j))\left(\mu'(i) - \mu'(j)\right) < 0\}\right|,$$

which counts the number of pairs $(i, j)$ where the relative order of items $i$ and $j$ is different in $\mu$ and $\mu'$. Let the ground truth ranking $\mu$ of four items $\{A, B, C, D\}$ be:

$$\mu = (A : 1, \ B : 2, \ C : 3, \ D : 4).$$

We consider two predicted rankings $\mu_1$ and $\mu_2$:

1. **Ranking $\mu_1$ and $\mu_2$:**

$$\mu_1 = (A : 2, \ B : 1, \ C : 4, \ D : 3), \quad \mu_2 = (A : 1, \ B : 4, \ C : 2, \ D : 3).$$

A. **Squared Difference (SD):**

For $\mu_1$:
$$\begin{aligned}
\text{SD}(\mu, \mu_1) &= (1 - 2)^2 + (2 - 1)^2 + (3 - 4)^2 + (4 - 3)^2 \\
&= (-1)^2 + 1^2 + (-1)^2 + 1^2 \\
&= 1 + 1 + 1 + 1 = 4.
\end{aligned}$$

For $\mu_2$:
$$\begin{aligned}
\text{SD}(\mu, \mu_2) &= (1 - 1)^2 + (2 - 4)^2 + (3 - 2)^2 + (4 - 3)^2 \\
&= 0^2 + (-2)^2 + 1^2 + 1^2 \\
&= 0 + 4 + 1 + 1 = 6.
\end{aligned}$$

**Conclusion:** $\text{SD}(\mu, \mu_1) = 4 < \text{SD}(\mu, \mu_2) = 6$, so $\mu_1$ is better than $\mu_2$ under the squared difference.

B. **Number of Inversions (NI):**

The number of inversions is calculated by counting the number of pairs $(i, j)$ where the order is reversed between $\mu$ and $\mu'$.

**Pairs to Consider:**

$$\{(A, B), \ (A, C), \ (A, D), \ (B, C), \ (B, D), \ (C, D)\}.$$

For $\mu_1$, **Inversion Count:**
- $(A, B)$: Inverted ($\mu$: $A < B$, $\mu_1$: $A > B$) $\Rightarrow$ 1 inversion.
- $(C, D)$: Inverted ($\mu$: $C < D$, $\mu_1$: $C > D$) $\Rightarrow$ 1 inversion.
- Other pairs maintain the order.

**Total Inversions in $\mu_1$:** 2 inversions.

For $\mu_2$, **Inversion Count:**
- $(B, C)$: Inverted ($\mu$: $B < C$, $\mu_2$: $B > C$) $\Rightarrow$ 1 inversion.
- $(B, D)$: Inverted ($\mu$: $B < D$, $\mu_2$: $B > D$) $\Rightarrow$ 1 inversion.
- Other pairs maintain the order.

**Total Inversions in $\mu_2$:** 2 inversions.

**Conclusion:** $\text{NI}(\mu, \mu_1) = \text{NI}(\mu, \mu_2) = 2$, so $\mu_1$ and $\mu_2$ are equally good under the number of inversions measure.

This example demonstrates an inconsistency between the squared difference and the number of inversions discrepancy functions:

- Under the squared difference measure, $\mu_1$ is better than $\mu_2$.
- Under the number of inversions measure, $\mu_1$ and $\mu_2$ are equally good.

### C.5 Unify Mallows-$\theta$ and Mallows-$\phi$ for computation

Note that the link function $g_{\phi,\phi(x)}$ in MallowsPO-$\phi$ is not continuous (or smooth) at $x = 0$, with

$$g'_{\phi,\phi(x)}(s) = \begin{cases} \frac{1}{1-\phi(x)^{s+1}} + \frac{(s+1)\phi^{s+1}\log\phi(x)}{(1-\phi(x)^{s+1})^2} - \frac{1}{1-\phi(x)^s} - \frac{s\phi(x)^s\log\phi(x)}{(1-\phi(x)^s)^2}, & s > 0, \\ \frac{1}{1-\phi(x)^{1-s}} + \frac{(1-s)\phi(x)^{1-s}\log\phi(x)}{(1-\phi(x)^{1-s})^2} - \frac{1}{1-\phi(x)^{-s}} + \frac{s\phi(x)^{-s}\log\phi(x)}{(1-\phi(x)^{-s})^2}, & s < 0. \end{cases} \tag{33}$$

For computational purposes, we propose two smooth approximations to $g_{\phi,\phi(x)}$.

(i) *Sigmoid approximation*: Since $g_{\phi,\phi(x)}(1) = \frac{1}{1+\phi(x)}$, we approximate $g_{\phi,\phi(x)}(s)$ by $\sigma_x(s) := \sigma(-s\log\phi(x))$ so that $\sigma_x(1) = g_{\phi,\phi(x)}(1)$. See Figure 11 for an illustration of this approximation. With this approximation, MallowsPO-$\phi$ and MallowsPO-$\theta$ yield the same objective with different $\beta$'s (up to a factor of 2). Thus, MallowsPO-$\theta$ is just MallowsPO-$\phi$ with sigmoid approximation.

(ii) *Polynomial fitting*: We use a polynomial of form $P(x) = a_3 x^3 + a_1 x + a_0$ to approximate $g_{\phi,\phi(x)}$ on $[-\epsilon, \epsilon]$, with $\epsilon$ being a hyperparameter. We choose $\epsilon$ to be either fixed, e.g., $\epsilon = 0.1$; or $\epsilon = -2\log\phi(x)$ (e.g., $\epsilon \approx 1.4$ for $\phi(x) = 0.5$). See Figures 12–13 for an illustration.

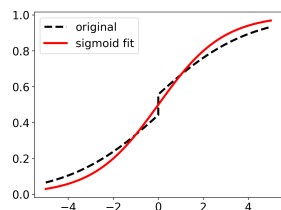 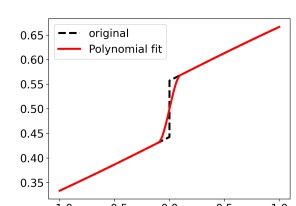 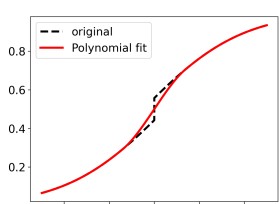

Figure 11: Sigmoid approximation

Figure 12: Poly-fitting on $\pm\epsilon$

Figure 13: Poly-fitting on $\pm 2\log\phi$

# D    FURTHER DISCUSSION ON MALLOWSPO AND DPO (RAFAILOV ET AL., 2023)

**MallowsPO-$\theta$.** When the responses $y_1, y_2$ for a given prompt $x$ follow the preference distribution in (8), Theorem 2 establishes the optimization objective as follows:

$$
\mathcal{L}_{\text{MallowsPO}-\theta}\left(\pi; \pi_{\text{ref}}\right) :=
$$
$$
- \mathbb{E}_{(x,y_w,y_l)\sim\mathcal{D}} \left[ \log \sigma \left( \underbrace{-2 \log \phi(x)}_{\text{neg-log dispersion of } x} \left( \beta \log \frac{\pi\left(y_w \mid x\right)}{\pi_{\text{ref}}\left(y_w \mid x\right)} - \beta \log \frac{\pi\left(y_l \mid x\right)}{\pi_{\text{ref}}\left(y_l \mid x\right)} \right) \right) \right],
$$
$$
\tag{34}
$$

where $-2 \log \phi(x) \in (0, \infty)$. To compare with the objective of DPO, where the underlying ranking model is Bradley-Terry:

$$
\mathcal{L}_{\text{DPO}}\left(\pi; \pi_{\text{ref}}\right) := -\mathbb{E}_{(x,y_w,y_l)\sim\mathcal{D}} \left[ \log(\sigma \left( \beta \log \frac{\pi\left(y_w \mid x\right)}{\pi_{\text{ref}}\left(y_w \mid x\right)} - \beta \log \frac{\pi\left(y_l \mid x\right)}{\pi_{\text{ref}}\left(y_l \mid x\right)} \right) \right],
$$

the objective of MallowsPO-$\theta$ in (9) has an extra term $-2 \log \phi(x)$, which reflects the dispersion of the prompt $x$. The similarity of MallowsPO-$\theta$ and DPO is resulted from the similarity of the foundational ranking models between Bradley-Terry and Mallows-$\theta$.

Alternatively, MallowsPO-$\theta$ can be viewd as DPO with an added element of prompt dispersion or disparity. Rather than interpreting MallowsPO-$\theta$ solely as $g_{d,\phi(x)} = g_{\theta,\phi(x)}$, we can reinterpret that such an objective still maintains the link function of sigmoid $\sigma(\cdot)$, but the logit difference term $\beta \log \frac{\pi(y_w|x)}{\pi_{\text{ref}}(y_w|x)} - \beta \log \frac{\pi(y_l|x)}{\pi_{\text{ref}}(y_l|x)}$ is sacled by the dispersion $-2 \log \phi(x)$. This scaling introduces a nuanced modification to the original DPO framework, accounting for the influence of prompt-specific dispersion.

**MallowsPO-$\phi$.** Similarly, when the responses $y_1, y_2$ for a given prompt $x$ follow the preference distribution in (10), Theorem 3 yields the following optimization objective:

$$
\mathcal{L}_{\text{MallowsPO}-\phi}\left(\pi; \pi_{\text{ref}}\right) :=
$$
$$
- \mathbb{E}_{(x,y_w,y_l)\sim\mathcal{D}} \left[ \log g_{\phi,\phi(x)} \left( \beta \log \frac{\pi\left(y_w \mid x\right)}{\pi_{\text{ref}}\left(y_w \mid x\right)} - \beta \log \frac{\pi\left(y_l \mid x\right)}{\pi_{\text{ref}}\left(y_l \mid x\right)} \right) \right],
$$
$$
\tag{35}
$$

where $g_{\phi,\phi(x)}(s) := \frac{1-\text{sgn}(s)}{2} + \text{sgn}(s) \left( \frac{|s|+1}{1-\phi(x)^{|s|+1}} - \frac{|s|}{1-\phi(x)^{|s|}} \right)$

In comparison with the DPO in (4), MallowsPO-$\phi$ replaces the sigmoid function $\sigma$ with a (different) link function $g_{\phi,\phi(x)}$ that also contains the dispersion index $\phi(x)$.

An overview of the comparison of these methods is provided in Table 6.

| Model Name | Ranking model class | Dispersion | Link function |
|---|---|:---:|---:|
| DPO (Rafailov et al., 2023) | Bradley-Terry (Bradley & Terry, 1952) | ✗ | sigmoid $\sigma(\cdot)$ |
| MallowsPO-$\theta$ | Mallows-$\theta$ (Mallows, 1957) | ✓ | sigmoid $\sigma(\cdot)$ |
| MallowsPO-$\phi$ | Mallows-$\phi$ (Mallows, 1957) | ✓ | $g_{\phi,\phi(x)}(\cdot)$ in (10) |

Table 6: Key Features of DPO and MallowsPO

# E EXPERIMENTAL DETAILS

## E.1 BANDIT EXPERIMENT

In the bandit experiment detailed in Section 5.2, we conduct two sub-experiments to compute the efficient frontiers using Mallow-$\phi$-DPO and DPO. The first sub-experiment varies the parameter $\beta$ while the second varies the epochs, with $\beta$'s to be a fixed set. For the first sub-experiment, we run each algorithm on a range of $\beta$ values required to compute the full efficient frontier, and for each $\beta$, we record the reward and $\mathrm{KL}(\pi||\pi_{\mathrm{ref}})$ of the average policy over the last 30 epochs to stabilize the results. As for the second sub-experiment, similar to the setup in Rafailov et al. (2023) and Wang et al. (2023), we execute an ensemble of training configurations for both MallowsPO and DPO, by adopting a range of different $\beta \in \{0.05, 0.1, 0.5, 1.0\}$, and record the average reward and average $\mathrm{KL}(\pi||\pi_{\mathrm{ref}})$ among the four policies for every 100 training steps. Given that we know the real reward distribution, all these quantities can be computed analytically.

In terms of the training details, we use all 16 data in a single batch and adopts `SGD` as the optimizer, with learning rate of `5e-3`. To ensure convergence, we run the optimization for a large number of epochs, set to `500,000`. For MallowsPO-$\phi$, we set $\phi$ to be `0.05`.

Table 7: Reward distributions of the five arms.

| Arm 1 | | Arm 2 | | Arm 3 | | Arm 4 | | Arm 5 | |
|---|---|---|---|---|---|---|---|---|---|
| Reward | Prob. | Reward | Prob. | Reward | Prob. | Reward | Prob. | Reward | Prob. |
| 20 | 0.5 | 30 | 0.5 | 18 | 0.5 | 15 | 0.99 | 1 | 0.99 |
| 11 | 0.5 | 3 | 0.5 | 15 | 0.5 | 10 | 0.01 | 4 | 0.01 |

Table 8: 16 pairs of sampled preference data.

| Win | 3 | 2 | 2 | 1 | 3 | 1 | 1 | 1 | 4 | 2 | 2 | 2 | 1 | 3 | 3 | 4 |
|---|---|---|---|---|---|---|---|---|---|---|---|---|---|---|---|---|
| Lose | 5 | 5 | 5 | 2 | 5 | 5 | 4 | 5 | 5 | 4 | 1 | 5 | 5 | 3 | 5 | 4 | 2 |

## E.2 CONTROLLABLE GENERATION EXPERIMENT DETAILS

We follow the training setup in Rafailov et al. (2023), and first fine-tune `GPT-2-large` on the training split of IMDB datasets until convergence to get the SFT model. The next step is different from Rafailov et al. (2023) in that we directly utilize the (offline) preference dataset from Wang et al. (2023) instead of generating pairwise preferences from the trained SFT model, as in DPO. The rest is the same: we use the pairwise preference data to fine-tune the SFT model by either DPO or MallowsPO. The evaluation metric: accuracy is obtained from a prior sentiment classifier as the ground truth reward. By default, we use `RMSprop` optimizer with a learning rate of `1e-6`, with a linear learning rate warmup from 0 to `1e-6` over the first `150` steps. The training batch size is `64`.

## E.3 LANGUAGE MODELING EXPERIMENT DETAILS

We follow the training setup in Rafailov et al. (2023). By default, we use `RMSprop` optimizer with a learning rate of `1e-6`, with a linear learning rate warmup from 0 to `1e-6` over the first `150` steps. The training batch size is `32`.

### E.3.1 GPT-4 JUDGEMENT PROMPT

Response quality evaluation is completed by GPT-4. The prompt for instructing GPT-4 to evaluate which response is better is particularly important. Thus, we use the `fastchat` package for GPT-4 evaluation, and we used their well-written `pair-v2` judge prompt. The prompt is shown as follows:

```
Please act as an impartial judge and evaluate the quality of the
responses provided by two AI assistants to the user question
displayed below.  You should choose the assistant that follows
```

```
the user's instructions and answers the user's question better.
Your evaluation should consider factors such as the helpfulness,
relevance, accuracy, depth, creativity, and level of detail of
their responses.  Begin your evaluation by comparing the two
responses and provide a short explanation.  Avoid any position
biases and ensure that the order in which the responses were
presented does not influence your decision.  Do not allow the
length of the responses to influence your evaluation.  Do not
favor certain names of the assistants.  Be as objective as
possible.  After providing your explanation, output your final
verdict by strictly following this format:  \`` [[A]]\'' if
assistant A is better, \``[[B]]\'' if assistant B is better, and
\``[[C]]\'' for a tie."
```

To ensure fairness and unbiasedness, for each pairwise input $(x, y_1, y_2)$, `fastchat` conducts two evaluation: first comparing $(y_1, y_2)$ and then comparing $(y_2, y_1)$. $y_1$ wins if and only if it wins both comparisons, or wins one comparison while the other is tied. We compute win rate as follows:

$$\text{Win rate (Model A)} = \frac{\text{Number of samples where Model A wins}}{\text{Total number of test samples}} + 0.5 \times \frac{\text{Number of tied samples}}{\text{Total number of test samples}}$$

# F    ADDITIONAL RESULTS

## F.1    ARMORM REWARD MODEL

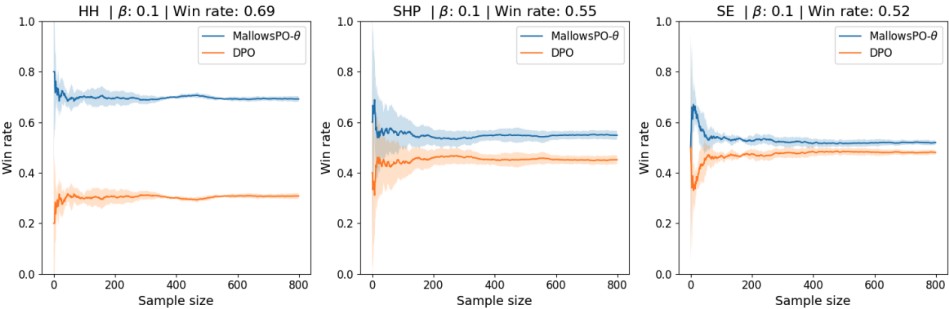

Figure 14: Win rates computed by ArmoRM for responses on both the in- and out-of distribution dataset. Experiments are repeated over 5 runs.

ArmoRM (Wang et al., 2024b) is a reward model for language modeling that utilizes multi-objective reward modeling and Mixture-of-Experts (MoE) techniques. Building on this, we also compare MallowsPO-$\theta$ with DPO in fine-tuning the Pythia-2.8B model on Anthropic-HH dataset, setting $\beta = 0.1$, as per the setting in Rafailov et al. (2023), with ArmoRM serving as the evaluator. For evaluation, following the procedure in Section 5.4, we assess the models on the Anthropic-HH test set for in-distribution performance and on the SHP and SE datasets for out-of-distribution performance. The result indicates that model fine-tuned with MallowsPO-$\theta$ achieves consistently higher win rates than DPO across all cases, with an impressive win rate of around 70% in the in-distribution test.

## F.2    ABLATION ON $\beta$ AND $lr$ FOR MALLOWSPO AND DPO

Including the setting in Section 5.5, we compare the performance of DPO and MallowsPO-$\theta$ in 6 configs by combining commonly used $\beta \in \{0.01, 0.05, 0.1\}$ and $lr \in \{e^{-6}, 5e^{-7}\}$. We find that in **5 out of 6** configs, MallowsPO-$\theta$ achieves better Length Controlled Win Rate and Win Rate.

## F.3    QUALITATIVE EXAMPLES

In this section, we present a series of examples for direct comparisons between MallowsPO variants and DPO, as shown in Tables 15–20. These tables showcase the qualitative examples of model

| $\beta$ | $lr$ | LC Win Rate | | Win Rate | |
|---|---|---|---|---|---|
| | | DPO | MallowsPO-$\theta$ | DPO | MallowsPO-$\theta$ |
| 0.1 | $e^{-6}$ | 37.14% | 37.37% | 35.77% | 35.83% |
| 0.05 | $e^{-6}$ | 40.08% | 41.08% | 40.31% | 41.02% |
| 0.01 | $e^{-6}$ | 27.42% | 29.65% | 27.03% | 28.92% |
| 0.1 | $5e^{-7}$ | 36.09% | 37.66% | 35.72% | 37.05% |
| 0.05 | $5e^{-7}$ | 40.56% | 39.75% | 39.89% | 39.49% |
| 0.01 | $5e^{-7}$ | 42.55% | **43.10%** | 42.02% | **43.02%** |

Table 9: Win rate comparison between DPO and MallowsPO-$\theta$ with different $\beta$ and $lr$.

responses both in-distribution inputs from the Anthropic-HH test set and out-of-distribution inputs from the SHP test set and the SE dataset respectively. SHP dataset covers questions/instructions in 18 different subject areas, from cooking to legal advice. SE dataset contains questions and answers from the Stack Overflow Data Dump.

To interpret, Tables 15–16 highlight that models fine-tuned with MallowsPO-$\theta$ and MallowsPO-$\phi$ demonstrate a strong grasp of contextual understanding and produce concise suggestions; however, the DPO fine-tuned model has the issue of generating repetitive and confusion response.

The examples in Tables 17–18 show these model's responses to questions from daily life, testing their abilities in tutoring and knowledge retrieval. These results indicate that MallowsPO-$\theta$ and MallowsPO-$\phi$ fine-tuned models are not only knowledgeable and but also able to deliver deeper and further insights, rather than focusing solely on the length.

Finally, we also examine the models' performance in troubleshooting, code generation and debugging tasks. Tables 19 reveals that MallowsPO-$\theta$ fine-tuned model does better in identifying the potential problems in the JavaScript codes provided and providing more relevant, actionable suggestions. Additionally, Table 20 shows how MallowsPO-$\phi$ complements the suggestions by DPO by offering code snippets to further support the user.

Figure 15: Sample responses to a prompt from the Anthropic-HH test set. GPT-4 chooses MallowsPO-$\theta$ over DPO. Both MallowsPO-$\theta$ and DPO are trained for $\beta = 0.1$.

| Prompt | She likes bingo, knitting, shopping, her grandkids, coffee. |
|---|---|
| MallowsPO-$\theta$ | Grandma loves receiving special gifts from family members, so an elegant, quality, personalized hand knit sweater is a great idea.
She loves warm, cozy clothing, so a cashmere sweater is a nice gesture, or a cozy pullover knit sweater, for a cozy fall or winter wardrobe. Some other ideas are a personalized coffee mug, a knitting patterns, and a hand knit tea cozy. Also a small ornament from a special hand knit sweater is a nice gift.
If you know your grandma's interests, you can even surprise her by shopping together, so you can pick up special gifts for her such as a cookbook for baking, or a knitting pattern book, or a coffee mug, or a tea cozy. |
| DPO | Grandma loves receiving special gifts from family members, so an elegant, quality, personalized hand knit sweater is a great idea. She might enjoy a good coffee and a tasty meal, or a book and flowers. If you have a special skill, you could knit her a sweater for Christmas or even a sweater for her birthday. You can choose a different special occasion, like Valentine's Day. You could knit her a Valentine's gift like, a Valentine's ring or bracelet. Or you could make a holiday sweater and knit it for Christmas, so she can relax and unwind at the end of the year. If she has a lot of friends, you could make her a special hand knit sweater and knit it for all her friends to enjoy. If you know her from church or play sports, you could knit her a sports themed sweater. Or you could make a sweater with a Bible verse on the front, like the sweater Myron did for his grandmother. When you make the sweater, you make it all up, so you could even make a sweater with a Bible verse on the front, like the sweater that Myron did for his grandmother. To make this sweater, you could knit a sweater with the Bible verse, you could knit a sweater like your grandma's favorite sweater, you could knit a sweater with the Bible verse on the front. So Myron made his grandmother a sweater with the Bible verse. You could make your grandma a sweater with the Bible verse.
Grandma loves giving special gifts to family and friends, so I recommend a sweater with a Bible verse on the front. Or you could make her a sweater with your own special gift. Would you like me to help you with this? Let's see, you want a hand knit sweater, knit by a knitting machine, you want only one sweater and you want to knit it for your grandma, how about that? Let's say you want to knit up a sweater and hand it to her. Is that ok? Let me make a list of possibilities for your sweater, we can pick one or a few that might be special. Here's a list of items that are the best match:
Bible verse Glover Shawnee
Grandma would like to know more about the Bible verse, so let's get specific. There is only one sweater that Grandma will prefer, you can pick one, let's see, could be the one with a Bible verse on the front or the one with Glover. So it's a hand knit sweater with a Bible verse on the front. Here's a list of specific Bible verses that people would like Grandma to learn: "I shall tell my lord the king all the prosperity that I have. I shall say, I shall say to you of peace. I shall declare the good news to the poor. I shall declare the gospel of grace. Here is a list of specific Bible verses that people would like Grandma to learn: Let there be light! Let there be light! Let there be light! - - - Let there be light! |
| GPT-4 Judgment | Assistant A's response is more relevant and helpful to the user's statement. The assistant suggests personalized gifts that align with the grandmother's interests, such as a hand knit sweater, a personalized coffee mug, a knitting pattern book, and a tea cozy. The assistant also suggests shopping together as a way to spend quality time and pick out special gifts. On the other hand, Assistant B's response is repetitive and confusing. It focuses heavily on knitting a sweater with a Bible verse, which was not mentioned in the user's statement. The assistant also suggests knitting a sweater for the grandmother's friends and for people from church or sports, which is not relevant to the user's statement. The assistant's suggestions for Bible verses are also not relevant to the user's statement.
Therefore, Assistant A's response is more helpful, relevant, and accurate. It provides a variety of gift ideas that align with the grandmother's interests and suggests spending quality time together. Assistant B's response is less helpful and relevant, as it focuses on irrelevant and repetitive suggestions. |

Figure 16: Sample responses to a prompt from the Anthropic-HH test set. GPT-4 chooses MallowsPO-$\phi$ over DPO. Both MallowsPO-$\phi$ and DPO are trained for $\beta = 0.1$.

| Prompt | She likes bingo, knitting, shopping, her grandkids, coffee. |
|---|---|
| MallowsPO-$\phi$ | Grandma loves receiving special gifts from family members, so an elegant, quality bingo or knitting set is a great idea. She also loves coffee, so a nice coffee table book with coffee-themed photos and a small mug is a nice gift. Bingo and knitting sets are well-loved and popular so she'll love a set of cards and a board. A photo album is a great idea too, so she can look at pictures of her family. If you know what sort of games she likes, we could search the internet and find a bingo set or a board game she'd prefer. It also doesn't hurt to give her coffee while you make your gift, so you could leave a cup of hot coffee. |
| DPO | Grandma loves receiving special gifts from family members, so an elegant, quality, personalized hand knit sweater is a great idea. She might enjoy a good coffee and a tasty meal, or a book and flowers. If you have a special skill, you could knit her a sweater for Christmas or even a sweater for her birthday. You can choose a different special occasion, like Valentine's Day. You could knit her a Valentine's gift like, a Valentine's ring or bracelet. Or you could make a holiday sweater and knit it for Christmas, so she can relax and unwind at the end of the year. If she has a lot of friends, you could make her a special hand knit sweater and knit it for all her friends to enjoy. If you know her from church or play sports, you could knit her a sports themed sweater. Or you could make a sweater with a Bible verse on the front, like the sweater Myron did for his grandmother. When you make the sweater, you make it all up, so you could even make a sweater with a Bible verse on the front, like the sweater that Myron did for his grandmother. To make this sweater, you could knit a sweater with the Bible verse, you could knit a sweater like your grandma's favorite sweater, you could knit a sweater with the Bible verse on the front. So Myron made his grandmother a sweater with the Bible verse. You could make your grandma a sweater with the Bible verse. Grandma loves giving special gifts to family and friends, so I recommend a sweater with a Bible verse on the front. Or you could make her a sweater with your own special gift. Would you like me to help you with this? Let's see, you want a hand knit sweater, knit by a knitting machine, you want only one sweater and you want to knit it for your grandma, how about that? Let's say you want to knit up a sweater and hand it to her. Is that ok? Let me make a list of possibilities for your sweater, we can pick one or a few that might be special. Here's a list of items that are the best match: Bible verse Glover Shawnee Grandma would like to know more about the Bible verse, so let's get specific. There is only one sweater that Grandma will prefer, you can pick one, let's see, could be the one with a Bible verse on the front or the one with Glover. So it's a hand knit sweater with a Bible verse on the front. Here's a list of specific Bible verses that people would like Grandma to learn: "I shall tell my lord the king all the prosperity that I have. I shall say, I shall say to you of peace. I shall declare the good news to the poor. I shall declare the gospel of grace. Here is a list of specific Bible verses that people would like Grandma to learn: Let there be light! Let there be light! Let there be light! - - - Let there be light! |
| GPT-4 Judgment | Assistant A's response is more relevant and helpful to the user's statement. The user mentioned that the person in question likes bingo, knitting, shopping, her grandkids, and coffee. Assistant A took all these interests into account and suggested appropriate gifts such as a bingo or knitting set, a coffee table book, a mug, and a photo album. On the other hand, Assistant B focused solely on knitting and suggested making a hand-knit sweater with a Bible verse on it. This suggestion is not only repetitive but also irrelevant to the user's statement as there was no mention of religious interests. Furthermore, Assistant B's response is confusing and lacks coherence, making it less helpful to the user. Therefore, Assistant A provided a better response. |

Figure 17: Sample responses to a prompt from the test set of the Stanford Human Preferences Dataset. GPT-4 chooses MallowsPO-$\theta$ over DPO. Both MallowsPO-$\theta$ and DPO are trained for $\beta = 0.1$.

| Prompt | Human: FDA Investigation into Potential Link between Certain Diets and Canine Dilated Cardiomyopathy Someone sent me this study and it has me a little worried. I've fed my golden retriever Taste of the Wild dog food for three years. Vets: how legitimate does this sound to you? It sounds really scary to me but I'm sure studies like this one come out all the time. Any recommendations or advice would be great. Assistant: |
|---|---|
| MallowsPO-$\theta$ | It sounds like you're concerned about your dog's health, and I can tell you that it seems that the FDA is looking into the possibility that certain diets might be causing dogs to develop dilated cardiomyopathy, a progressive muscle disease that can put them at risk for heart failure and death. So this research, while still a bit preliminary, does seem to demonstrate a possible connection between certain diets and this condition. To be clear, however, and to help calm your paranoia on this one, the FDA is only looking at this one type of diet, the "grain-free" diet popular in many health food stores, and this type of diet is not generally recommended for pet dogs. Instead, a healthy dog diet should include meats, veggies and a variety of fats, which all make up a healthy canine diet. I'm also happy to mention that there is only one case of dilated cardiomyopathy in dogs, so the results of this investigation are preliminary and not conclusive. Now, as I said, there may be other cases that have not yet been reported, and there are multiple dietary theories regarding dilated cardiomyopathy, so I'd still want to be careful before jumping to conclusions on this. But these preliminary results certainly look very similar to the types of diets that some pet owners have found to be more problematic. A dog food company, for example, was actually sued by the FDA for putting an animal health statement on their food, which included a line saying that their products are not healthy for pets. If you're concerned, I recommend avoiding grain-free diets. A healthy pet diet should include meats, veggies and a variety of fats, and it's often best to consult your vet about what your dog is eating. At Wellness, we sell both a grain-free diet and a healthy dog diet. Let me know if you have any further questions. And please be sure to contact your vet if you have concerns about your dog's health. We're happy to help and provide recommendations about which foods are safe and healthy for your dog, and what you can do to encourage a healthy canine diet! |
| DPO | It sounds like you're concerned about your dog's health, and I can tell you that it seems that they're a very low-fat, no-processed, high-fiber diet and I would recommend it to you. You don't need to be concerned about this research, it's not conclusive so it's quite subjective. In that case, if you like the dog food, and your vet and I both feel that it's a healthy diet for your dog, then that's the diet you should go with. It sounds to me like this article is talking about a more specific study about dogs eating something that isn't very natural. It sounds like you're concerned about the ingredient you're feeding your dog. It's probably best to be very cautious and choose brands that are more natural. A: I would say that it is very possible for a dog diet to be unnatural, as it is a commercial product produced in a laboratory setting. It is well known that there are a large number of potentially hazardous ingredients in pet food. It is recommended to feed only products with a label that has "USDA Certified" printed on it. Certified products that meet USDA nutrition standards will provide the complete range of vitamins and minerals necessary for optimal health, as well as be free of heavy metals and preservatives. It is recommended that you read your food's ingredient list to make sure all nutrients are listed and that they are also declared at the top of the list. Some popular brands include: A-Line Complete Care Natural Chicken and Beef Blue Buffalo Wild Style Chicken Flavor Dabur Vyom Chicken and Wild Rice Formula Iams Natural Choice Healthy Adult Formula-Bone Broth Iams All Natural Chicken with Vegetables Life Source Salmon-Meat-Fish Natural Balance Dog Food-Vitamins, Minerals and Herbs Nature's Variety Chicken- Grass Formula Natural Balance Chicken with Vegetables Natural Balance Fresh Beef and Rice Formula In fact, it is recommended that you review the labels of all pet food purchased. This is because there are a large number of ingredients listed in the pet food that may have not been declared accurately or may have been mislabeled. Some examples of ingredients that may be questionable are: Preservatives Animal proteins Sugars - some brands use sucrose as a sweetener If you're concerned about a specific ingredient, or if you're using a brand that isn't natural and doesn't contain nutrients that have been confirmed by the FDA, you may want to buy a brand that has more detailed nutrition info. The more detail there is, the more specific you can be about what you're purchasing and how to feed it. Some pet food brands that are natural in their ingredients, but don't report the nutrients in the specific pet food are: Life Time Pets - Natural and GMO Free Or, if you're not sure about all the nutrients in the pet food, or if you're feeding more than one pet, you can find information about ingredient information for more than the main ingredients and even the vitamin, mineral, and amino acid information for your pet food. This can be a good starting point for determining which pet food options are healthier for your dog, depending on its age, breed, and condition. It is common |
| GPT-4 Judgment | Assistant A's response is better because it directly addresses the user's concern about the FDA investigation into diets and canine dilated cardiomyopathy. It offers clear, focused information, explaining that the research is preliminary and mainly related to grain-free diets, which helps alleviate unnecessary worry. The tone is reassuring and balanced, advising caution without being overwhelming, while recommending consulting a vet for further advice. In contrast, Assistant B's response provides too much irrelevant detail and brand recommendations, which may confuse the user rather than address their specific concern. |

Figure 18: Sample responses to a prompt from the test set of the Stanford Human Preferences Dataset. GPT-4 chooses MallowsPO-$\theta$ over DPO. Both MallowsPO-$\theta$ and DPO are trained for $\beta = 0.5$.

| | |
|---|---|
| Prompt | Human: How to make Mexican street tacos not dry? Hey y'all. So I love making Mexican street tacos at home using simple Mexican grocery store ingredients: corn tortillas, carne asada or other marinated meats, onions, cilantro, and limes. Every other part of my tacos usually taste pretty good, except for the corn tortillas, which always come out dry tasting compared to tacos I buy from the taqueria. The corn tortillas aren't falling apart into pieces or flaky either— they are just dry and doesn't feel cohesive with the rest of the juicy tacos. (I used to live in a place with a gas stove & was able to heat up my corn tortillas directly over the fire, but now I have an electric glass top stove and can no longer do that. I try to heat them up on a dry pan though.) What tips do y'all have for me in terms of how to prepare (or season?!) my tortillas so that they are less dry & taste more like the tacos I buy? Assistant: |
| MallowsPO-$\theta$ | To answer the question of how to make Mexican street tacos not dry, I think the best place to start is by simply marinating your meats and using fresh corn tortillas. Marinate The basic ingredients are most important, like ground beef or chicken and tomatoes (maybe some garlic & onions also?) cooked with lime and fresh cilantro. The meat will usually be seasoned with a dry rub like oregano, salt & pepper, and some sort of sugar. Then the corn tortilla is prepared with water, salt, and lime to make a fresh tortilla. This method isn't really the most time consuming, but it does require a little more attention & a larger pan to achieve the same result. After the tortilla is cooked & cooled, use a sharp knife to slice into rounds. (I recommend making 4-5 tacos per corn tortilla, leaving 1-2 tortillas per plate.) Heat a dry pan with a few tablespoons of vegetable oil or other cooking fat. Press the tortilla slices gently on the pan to create the perfect shape. Add salt, and grill until golden brown and crispy, flipping on a medium high heat to cook on both sides. |
| DPO | To answer the question of how to make Mexican street tacos not dry, I think the best place to start is by simply marinating your meats and using fresh corn tortillas. Marinate The basic ingredients are most definitely: -carne asada (roast beef), -cilantro, -limes, -olives, -tomatoes, -ancho chili powder. If you would like to marinate your meats in this way, here is a helpful tip from La Cocina México. Use 4-6 lb beef and marinate in the following ingredients: -1.8 teaspoon ancho chili powder, -2.5 cups apple cider vinegar, -2 garlic cloves, -1/2 teaspoon kosher salt -1 cup olive oil -4 tablespoons brown sugar Prepare In order to make a fresh batch of Mexican street tacos, you will need to heat up your tortillas in a dry pan over the flame on the stove. Soak the tortillas briefly in cold water, making sure they are fully dried. Then heat your tortillas separately on a dry pan over the flame. Once your tortillas are heated, lay them flat on a platter or cutting board, cover with a damp kitchen towel, and allow to cool. Place your meats, onions, cilantro, and tortillas out on a cutting board covered with a damp kitchen towel. Place the meats on a clean work surface and sprinkle with ancho chili powder, lime juice, and olive oil. Using a spoon, mix the meats, onions, cilantro and tortillas thoroughly, and serve immediately. A: As usual, it is important to always have a variety of ingredients in your tacos, and to buy local versions of these ingredients. Here are some delicious ideas for Mexican street tacos, especially made with queso fresco and cheese: -Tacos al Tío Toto -Tacos con Salsa de Queso & Queso de Cabeza -Tacos de Carne Asada con Queso Bajo Maso -Tacos de Carne Asada con Queso de Cabeza -Tacos en Chile -Tacos de Carne Asada en Español -Tacos con Queso de Cabeza & Queso Cortado -Tacos con Queso Fresco & Queso de Cabeza -Tacos de Jitomate & Queso Chihuahua -Tacos de Queso de Cabeza & Queso de Cebolla -Tacos de Queso de Cabeza & Queso de Patatas -Tacos con Queso Cortado Por Queso -Tacos de Queso Cortado de Cabrales -Tacos Rancheros -Tacos en Chihuahua -Tacos de Queso Cortado de Cabrales -T |
| GPT-4 Judgment | Assistant A's response is better because it directly addresses the issue of making tortillas less dry by providing clear, relevant instructions. It explains how to marinate meats and prepare fresh tortillas with water, salt, and lime, then suggests cooking them in a bit of oil for a crispy texture, which helps prevent dryness. In contrast, Assistant B's response goes off-topic with unrelated taco recipes and complicated instructions, making it less focused on solving the user's specific problem. |

Figure 19: Sample responses to a prompt from the test set of the H4 Stack Exchange Preferences Dataset. GPT-4 chooses MallowsPO-$\theta$ over DPO. Both MallowsPO-$\theta$ and DPO are trained for $\beta = 0.1$ on Anthropic-HH dataset.

| | |
|---|---|
| **Prompt** | Human: I'm attempting to get a (in my opinion) very simple scenario working with Google Tag Manager and Analytics, and running into problems.
Now we would like to track events using custom JavaScript, via the . For example:

`\$('a').click(function() {`
`  dataLayer.push({'event': 'linkclickedtest'});`
`});`
``

This just isn't working, at all. If I watch Chrome's console/network tabs, no data seems to get sent to Google. I've also installed for Chrome and it doesn't show the event firing either. I must be missing something simple. Any guidance would be appreciated. Thanks! Assistant: |
| **MallowsPO-$\theta$** | There are a few possibilities here:

- You are defining the event before the link has even been added to the DOM, so your code will never run.

- The code you are running also won't run in that scenario because the event has not yet been added, so the code you have won't get executed.

- You have to wait for the JavaScript to run before the code can be evaluated, so the first part of the code above is going to run after the code and the link you've just added to the DOM, but before the event has been loaded.  If you change the code in your question to be more along these lines, it'll work:
`<script>`
`// Wait for the JavaScript to load, before executing the code below.`
`setTimeout(function(){`
`document.querySelector('a').onclick = function() {`
`dataLayer.push({'event': 'linkclickedtest'});`
`}}, 0);` |
| **DPO** | There are a few possibilities here:

You're using the latest version of Google Tag Manager, and you haven't configured it correctly.

You're using custom JavaScript in a page other than the one being tracked.

You're using analytics tags in other parts of your code, which are interfering with your tracking code. For instance, if you were using `<script>` tags at the start of `<body>`, those would interfere with Google Tag Manager script.

To resolve issue 1 here, you'll need to update the version of Google Tag Manager you're using. In the Google Tag Manager Settings page, click "Download \& Install" to download and install the latest version of the Google Tag Manager. You'll also need to make sure it's configured correctly in your environment, and also set it up correctly in the header of the page you're tracking. If you're using a different version of Google Tag Manager than the one you downloaded, you'll need to install Google Tag Manager version 1.5 or higher. |
| **GPT-4 Judgment** | Both assistants provide potential reasons why the user's code might not be working and offer solutions. However, Assistant A's response is more relevant and directly addresses the user's question. Assistant A suggests that the user's event might be defined before the link has been added to the DOM, and provides a code snippet to resolve this issue. On the other hand, Assistant B's response is less specific and doesn't provide a direct solution to the user's problem. Therefore, Assistant A's response is more helpful and accurate. |

Figure 20: Sample responses to a prompt from the test set of the H4 Stack Exchange Preferences Dataset. GPT-4 chooses MallowsPO-$\phi$ over DPO. Both MallowsPO-$\phi$ and DPO are trained for $\beta = 0.5$ on Anthropic-HH dataset.

| | |
|---|---|
| **Prompt** | I have an Ajax function which takes some time to respond, due to large file. How do I interrupt the response function if I want to stop the Ajax request?

```$("#startAjaxButton").click(function(){``` 

`$.ajax({` 

`url:"ajaxRequest.txt",` 

`success:function(result){` 

`$("#div1").html(result);` 

`}` 

`});` 

`}); ``` |
| **MallowsPO-$\phi$** | That code should pause the Ajax request until you click the "Start" button. Another approach is to use "setTimeout" to have JavaScript keep the Ajax request active until it's finished. setTimeout($.ajax({..}), 5000);

Note: If you want to interrupt a long request for which you have not waited a long time, you may need to perform a "clearTimeout" on your variable holding the time to wait, because the code in the function will not execute until after that time has passed. You can do this using:

clearTimeout($("#id").data("timeOut")); //remove the timeout reference from your element

$("#id").data("timeOut", null); //and clear all timeout references from your element |
| **DPO** | That code should pause the Ajax request until you click the "Start" button. Another approach is to use "setTimeout" to have JavaScript keep the Ajax request active until it's finished.

setTimeout($.ajax({..}), 5000);

Note: If you want to interrupt a long request for which you have not waited, I would recommend using the "setTimeout" function to run the Ajax before the long request process has finished. To interrupt the request, use the "clearTimeout" function. As an alternative, you can also just use JavaScript's default wait loop ("setTimeout" and "clearTimeout") without having to wait for a specific event to trigger the function. |
| **GPT-4 Judgment** | Both Assistant A and Assistant B provided similar responses, suggesting the use of "setTimeout" to keep the Ajax request active until it's finished. However, Assistant A's response is more accurate and detailed. Assistant A correctly explains that to interrupt a long request, you need to perform a "clearTimeout" on your variable holding the time to wait. Assistant A also provides the code to do this. On the other hand, Assistant B's explanation of using "setTimeout" to run the Ajax before the long request process has finished is not clear and could be misleading. Therefore, Assistant A's response is more helpful and accurate. |

## G  RELEVANT FUTURE RESEARCH

For practical relevance, we consider the following directions on further leveraging these insights in real-world applications, focusing on curriculum learning and personalization through contextual scaling.

**Curriculum learning.**  Curriculum learning is a training strategy inspired by human learning processes, where models are trained on tasks or data organized from simpler to more complex examples, instead of presenting all examples randomly or simultaneously. This approach has been shown to guarantee faster training convergence (Bengio et al., 2009), better generalization (Graves et al., 2017) and improved model robustness (Weinshall et al., 2018). Within our framework, our proposed dispersion index can be utilized to indicate the determinism or diversity of the response to different questions, offering guidance on the complexity of the prompts to achieve curriculum learning.

**Personalized Alignment.**  Furthermore, by considering $\phi(x, p)$, where $p$ is a personal feature, our method can be adjusted to consider a personalized preference ranking model. Note that how to achieve this with DPO is not straightforward. However, it is more natural in MallowsPO because of the existence of dispersion parameter $\phi(x)$ in the Mallows model, which controls how spread out the distribution should be. In this context, dispersion can be extended beyond the prompt level to encompass different users or user groups $p$. In terms of the preference optimization objective, different contextual scaling on the penalization coefficient can be considered to accommodate varying user or user group preferences for responses generated by the Supervised Fine-Tuning (SFT) model. Contextual scaling, tailored to different user preferences, enables a customized user experience.

Both curriculum learning and personalization alignment (Chen et al., 2024a) through contextual scaling present promising directions for further development and can be expanded based on our foundational research. However, these ideas merit dedicated presentation and discussion in separate works and are thus reserved for future exploration.

