# OpenReview forum: "MallowsPO: Fine-Tune Your LLM with Preference Dispersions"
_ICLR.cc/2025/Conference — ICLR 2025 Poster_

### Official Review · Reviewer_8pkH · 2024-10-21

**Soundness:** 2
**Presentation:** 2
**Contribution:** 2
**Rating:** 5
**Confidence:** 4

**Summary:**

This paper proposes MallowsPO, which is based on DPO but is distinct in a few points, to fine-tune LLMs from human feedback. The main two distinctions are as follows; (1) Instead of assuming that the preference between the two sentences $y_1$ and $y_2$ is sampled from some probability distribution, the proposed approach assumes that the ranking ($\mu$) of $n$ items are sampled from distribution, and (2) assuming that the preference has some disparity depending on the input text. The proposed MallowsPO formulates a DPO-like objective function for the ranking assumption while incorporating the disparity by introducing a disparity parameter in the reward function.

**Strengths:**

- **Formulating disparity in preferences**: while the original preference model assumes the same preference distribution across different input sentences ($x$), this paper spotlights the fact that the preference distribution can change depending on ($x$) and formulate the difference in reward function.

- **Better empirical results by formulating disparity**: It seems that the result suggests that considering the disparity can improve the reward for varying values of KL divergence.

**Weaknesses:**

- **Assuming the probability distribution of ranking, rather than the pairwise preference, loses information about reward difference.**

The paper considers the probability distribution in ranking from the similarity between the ranking $\mu$ and so-called "central" ranking $\mu_0$. For example, Mallow-$\theta$ models the probability as a function of Spearman's rank difference to the central ranking. Unfortunately, this formulation loses the performance difference between each item.

For example, consider the case where we have three sentences, $(y_1, y_2, y_3)$. Also suppose that the reward of each is $(100, 1, 0)$. In this case, under the assumption of the paper, the probability of observing the ranking of $(100, 0, 1)$ and $(1, 100, 0)$ is the same, while the former is definitely more likely to be observed given the reward difference. For this reason, the proposed approach loses the information about the reward, and I could not find any advantage of assuming probability distribution on ranking rather than on pair-wise preference (pair-wise preference already implicitly encodes the probability of ranking, without losing the information about rewards).

- **Insufficient ablations: lacking baseline of <DPO+disparity>**

The main motivation of this paper is to take the disparity, or the difference of preference distribution depending on the inputs $x$ into account. Then, why isn't simply considering the disparity in the formulation of DPO sufficient? That is, because the preference distribution of DPO is assumed as

$p^{\ast}(y_1 \succ y_2 | x) := \sigma(r^{\ast}(x, y_1) - r^{\ast}(x, y_2))$, where $\sigma(z) = 1 / (1 + \exp(-z))$,

I think using $\phi(x) \cdot z$ (where $\phi(x)$ is the estimated disparity params) as the input of $\sigma(\cdot)$ is enough to consider the disparity. This simple baseline is not compared in the experiments.

---

**(summary)**
Overall, I did not understand why this paper needed to introduce the ranking formulation, which actually seems less beneficial than DPO. The latter, considering the disparity can be a unique contribution to this paper, however, this point seems somewhat incremental and weak as the ICLR paper. This is the reason for my evaluation.

**Questions:**

[questions]
- For Figure 9, how does the training curve differ for different prompts, i.e., $x$? It may be interesting to see how the action choice probability can change with the estimated disparity parameters ($\phi(x)$).


[nits]
- The label of Figure 8 (b) is maybe a mistake: I think it should be # of epochs rather than KL.

---

> ### Author Response · Authors · 2024-11-16
> **Clarification on Mallows Ranking Model**
>
> Thank you for providing suggestions to better improve the paper. We address your questions below:
>
> > Assuming the probability distribution of ranking, rather than the pairwise preference, loses information about reward difference.
>
> We understand the reviewer's concern and agree with the simple example. However, we’d like to point out that this concern, in the setting of language modeling, does not matter for several reasons.
>
> * Firstly, the reviewer’s concern raised about no difference in the probability of the rankings by swapping any two items will be resolved as the total number of items to rank are sufficiently large—a scenario that is common in language modeling tasks. Here we illustrate the effect of additional items for the relative probability of observing two rankings which differ only in the rankings of 3 items. If there are no additional items, then observing a relative order of items of value 100,2,1 in 100,1,2 in Mallows-$\theta$ is (i.e. observe (1,3,2) for central ranking (1,2,3)):
> $$
> p([100,1,2]) = Z\cdot \phi^{(3-2)^2+(2-3)^2} = Z\cdot \phi^2,
> $$
> which will be the same as observing (2,100,1) since:
> $$
> p([2,100,1]) = Z\cdot \phi^{(2-1)^2+(1-2)^2} = Z\cdot \phi^2.
> $$
> However, if there are additional 97 items of value (100 items in total of value from 1 to 100), things will be quite different. Consider the central ranking is  (100,99,98,97,...,2,1) for the items of value from 1 to 100. If two observed ranking are $\mu_1$ and $\mu_2$, and $\mu_1(i)=\mu_2(i)=100-i$ holds for all other $i$'s from 3 to 99, we have (100,1,2) becomes (100,99,98,...,3,1,2) and (2,100,1) becomes (2,99,98,...,3,100,1). Thus:
> $$
> p([100,99,98,97,\cdots,3,1,2]) = Z\cdot \phi^{(1-2)^2+(2-1)^2} = Z\cdot \phi^2,
> $$
> while
> $$
> p([2,99,98,97,\cdots,3,100,1]) = Z\cdot \phi^{(2-100)^2+(100-2)^2} = Z\cdot \phi^{2\times 98^2},
> $$
> which means two observations $p([2,100,1])<<p([100,1,2])$ since $\phi\in (0,1)$. We also included such a discussion in the updated version of our draft, please see **Appendix C.3**.
>
> * Secondly, in DPO’s formulation, there is no explicit step for reward modeling, which is a core strength as well as the beauty of the approach – its simplicity and directness are preserved by excluding the reward modeling step. Therefore, although for MallowsPO in theory there is an underlying ranking function, in practice the final objective of MallowsPO similarly does not involve its modeling either. Moreover, from this perspective, our MallowsPO framework embraces DPO as a special instance of MallowsPO-\theta, by letting $\phi(x) = 1$. Thus, in the context of preference optimization, MallowsPO provides a more comprehensive representation of information compared to DPO, ensuring that valuable insights are preserved rather than diminished.
>
> * Lastly, from the perspective of human feedback, it is more appropriate and natural to model the rank of an object rather than the reward. Consider the following toy example: given two articles A and B, it is generally easier for a person to tell their preference between the two rather than assign precise scores to each article. In the original technical report of InstructGPT [2], human annotators provided preference data by ranking sets of 4 to 9 responses instead of assigning exact reward scores, illustrating this principle.
>
> ---
>
> **Reference**
>
> [1] Mallows, C. L. (1957). Non-null ranking models. I. Biometrika, 44(1/2), 114-130.
>
> [2] Ouyang, L., Wu, J., Jiang, X., Almeida, D., Wainwright, C., Mishkin, P., ... & Lowe, R. (2022). Training language models to follow instructions with human feedback. Advances in neural information processing systems, 35, 27730-27744.

---

> ### Author Response · Authors · 2024-11-16
> **Equivalence of <DPO+disparity> and MallowsPO-$\theta$**
>
> > Insufficient ablations: lacking baseline of <DPO+disparity>
>
> Long story short, **the proposed <DPO+disparity> is exactly MallowsPO-$\theta$**, one of the MallowsPO instances, please see Corollary 2 of Section 3.2. The proposed method is simple yet novel while offering valuable insights.
>
> * Firstly, we’d like to again emphasize that this method is not proposed as an isolated and ungrounded method but from the pairwise probability of the Mallows-$\theta$ model. This foundational link ensures the robustness and theoretical basis of our method.
>
> * Moreover, to recall one of our contributions, we address the issue of how to efficiently estimate the dispersion or disparity parameter using the entropy associated with winning and losing items. This idea builds on the interpretation of the dispersion parameter in the Mallows model, which quantifies distribution spread and is conceptually related to entropy.
>
> * Thirdly, DPO + disparity—also referred to as MallowsPO-$\theta$ in our work—represents just one instance within the broader MallowsPO framework, which is theoretically rooted in the Mallows model. As demonstrated in Theorem 1, we establish a one-to-one mapping between the distance function in the Mallows model and the MallowsPO loss objective. This highlights that numerous other instances of MallowsPO can also be derived and explored.
>
> * Lastly, the idea of contextual scaling also originates from the MallowsPO framework. Specifically, it corresponds to the term $-2\log\phi(x)$ in Corollary 2 (multiplied by regularization parameter $\beta$ and logits difference), motivating us to use MallowsPO as a modular component for enhancing other preference optimization methods, such as IPO and simPO, to achieve further performance boosting.

---

> ### Author Response · Authors · 2024-11-16
> **Other Questions and Comments**
>
> > For Figure 9, how does the training curve differ for different prompts, i.e., $x$?
>
> As stated in line 404, there is no prompt or contextual information $x$ for the bandit experiment, as it involves selecting from a predetermined set of arms. Without contextual information $x$ to estimate the dispersion, the dispersion parameter is fixed, in our case at 0.05, for illustration.
>
> > The label of Figure 8 (b) is maybe a mistake: I think it should be # of epochs rather than KL.
>
> Thank for your question of clarification and we want to say that this is not a mistake. We meant to plot KL vs reward because it demonstrates the dynamics of the reward of the training policy and its KL with respect to the reference policy for every 100 epochs during training. KL regularization has already been extensively employed in existing literature, either as a practical computational constraint [3][4] or as a mechanism to balance the trade-off between accuracy and diversity [5].
>
> To interpret the plot in Figure 8(b), at the beginning, the training policies of both methods are identical to the reference policy, resulting in 0 KL and around 12.9 reward. As the training progresses, the KLs of the training policies for both methods will become larger and larger. As a result, the training policy of DPO converges to the one with larger KL and smaller reward.
>
> ---
> **References**
>
> [3] Tang, Y., Guo, Z. D., Zheng, Z., Calandriello, D., Munos, R., Rowland, M., ... & Piot, B. (2024). Generalized preference optimization: A unified approach to offline alignment. ICML 2024
>
> [4] Munos, R., Valko, M., Calandriello, D., Azar, M. G., Rowland, M., Guo, Z. D., ... & Piot, B. (2023). Nash learning from human feedback. ICML 2024
>
> [5] Wang, C., Jiang, Y., Yang, C., Liu, H., & Chen, Y. (2023). Beyond reverse kl: Generalizing direct preference optimization with diverse divergence constraints. ICLR 2024.

---

> > ### Comment · Reviewer_8pkH · 2024-11-22
> >
> > Thank you for the detailed responses.
> >
> > **the proposed <DPO+disparity> is exactly MallowsPO-$\theta$**
> >
> > Thank you for showing the connection, but I'm confused with this explanation. DPO aims to fit the sentence probability as $\pi(y | x) \propto \sum_y  \pi_{\text{ref}}(y | x) \exp (r(x, y) / \beta)$, while MallowsPO-$\theta$ aims to fit the ranking probability as $P(\mu | x) \propto \phi^{d(\mu, \mu_0)}$. How do these become equivalent to each other? Especially, in the simple case with [100, 2, 1], I do not understand how MallowsPO-$\theta$ becomes equivalent to DPO, while having the same probability on [100, 1, 2] and [2, 100, 1]. Also, can you elaborate the derivation of Eq. (8)? From the current draft, I think it is hard for readers to understand that MallowsPO-$\theta$ will be equivalent to <DPO+disparity>.

---

> ### Author Response · Authors · 2024-11-23
>
> Much thanks for your reply and for asking further clarification questions.
>
> > How do <DPO+disparity> and MallowsPO-$\theta$ become equivalent to each other?
>
> We would like to showcase the equivalence of these two in terms of their resulting **loss objectives**. For the same preference dataset, both DPO and MallowsPO-$\theta$ end up with a loss objective to optimize, thus we could take their loss objectives as our first class citizens.
> DPO loss (as in Equation (4)) is derived by combining the cross-entropy loss and the change of variable technique (as you mentioned):
> $$
> -\mathbb{E}\log \sigma \left(r^{\theta}(x,y^w)-r^{\theta}(x,y^l)\right)
> $$
> where the expectation is with respect to the preference dataset $\mathcal{D}=(x,y^w,y^l)$ and $r^{\theta}(x,y) = \beta \log \frac{\pi_{\theta}(y \mid x)}{\pi_{\text {ref }}(y \mid x)}$ is known as the implicit reward model of DPO. Compared to DPO loss, MallowsPO-$\theta$ loss objective differs only in an extra contextual term which corresponds to our dispersion estimators (we write as $f(x)$ for simplicity):
> $$
> -\mathbb{E}\log \sigma \left(f(x)\left(r^{\theta}(x,y^w)-r^{\theta}(x,y^l)\right)\right).
> $$
> This is exactly <DPO+disparity>  (if calling $f(x)$ as the disparity term, while in our paper we refer as dispersion term), that's what we mean by the **equivalence of MallowsPO-$\theta$ and DPO**. We also want to emphasize that ultimate DPO "training" **does not involve any exact reward** as in the loss objective, and only relative order/rank matters. Similarly **the exact rank does not appear** in the objective function of MallowsPO. So whether the true underlying reward is (100, 1, 2) or (2024,1,2) will not change the DPO or MallowsPO loss objective. To capture the significance of DPO preference pairs to improve DPO has been explored in papers like ODPO [1], but this is beyond the scope of our work.
>
> > Also, can you elaborate the derivation of Eq. (8)?
>
> Thanks for your suggestion. We provide a more detailed derivation of the Equation 8 in Corollary 2 in the Appendix B and mark it in blue for your reference. We also include the concrete objective of MallowsPO-$\theta$ and MallowsPO-$\phi$ in Appendix B and Appendix G to showcase a more clear connection between MallowsPO-$\theta$ and <DPO+disparity>. Briefly speaking, Equation (8) illustrates that pairwise preference is indeed Bradley-Terry with each item’s reward $y$ being $\log(\phi(x)) \mu_0(y\mid x)$, with $\mu_0$ denoting the central ranking.This further justifies what we claim about why MallowsPO-$\theta$ can be understood as DPO+disparity.
>
> ---
> **References**.
>
> [1] Direct Preference Optimization with an Offset (ODPO) https://arxiv.org/pdf/2402.10571

---

> > ### Comment · Reviewer_8pkH · 2024-11-25
> >
> > Thank you for the response. Additional clarification in the Appendix seems reasonable, and I will raise my score to 5.
> >
> > This is independent of the evaluation score, but MallowsPO-$\theta$ and DPO are still not exactly the same, right? I think MallowsPO-$\theta$ corresponds to DPO with some surrogate reward (which is different from the reward definition of DPO), because under the formulation of MallowsPO-$\theta, reward = [100, 2, 1] and reward = [50, 2, 1] should have the same ranking probability under the model described in Section 3.1. It would be informative for the readers to clarify these differences and also include <DPO + sparsity> as a additional baseline taking this point into account.

---

> > > ### Author Response · Authors · 2024-11-25
> > >
> > > Thank you for raising the score and for your thoughtful feedback! We truly appreciate your interest in our work and would like to provide further clarifications.
> > >
> > > > MallowsPO-$\theta$ and DPO are still not exactly the same, right?
> > >
> > > Indeed MallowsPO-$\theta$ and DPO are different no matter from the perspectives of loss objective or ranking model. In terms of loss objective, MallowsPO-$\theta$ considers to weigh the logit different term additionally by the dispersion term $-2\log\phi(x)$ while DPO does not. In terms of ranking model, MallowsPO-$\theta$ relies on Mallows-$\theta$ while DPO relies on Bradley-Terry.
> > >
> > > However, MallowsPO-$\theta$ and DPO+disparity (assuming the reviewer refers DPO+disparity to the form of $E[-\log \sigma (-2\log\phi(x) \cdot \text{logit difference})]$, but if not, please feel free to correct too), are equivalent in the sense of loss objective. We still would like to emphasize that DPO (or MallowsPO) does not involve a reward model during training. So the concern of the reviewer is more related to the theoretical perspective rather than how the method works in reality, as the training is solely driven by the data and the loss objective.
> > >
> > > >  MallowsPO-$\theta$ corresponds to DPO with some surrogate reward
> > >
> > > In terms of the ranking model, we want to further clarify that there is not a reward function in Mallows ranking model, but we acknowledge the reviewer’s interpretation in the sense that the pairwise probability of Mallows-theta can be viewed as Bradley-Terry with some form of reward, which is $\mu_0(i) \cdot \log \phi$, where $\mu_0(i)$ is the central rank of ith item. However, as we mentioned before, in a typical offline RLHF setting, we only observe preferred and not preferred responses to the prompts, thus modeling the ranks of the responses is already sufficient. Secondly, we would like to mention again that there is no reward modeling in DPO.
> > >
> > > We hope this clarification addresses the reviewer’s concerns, and provides insight into the distinctions and underlying mechanics of MallowsPO-$\theta$ and DPO, and the equivalence between MallowsPO-$\theta$ and DPO+disparity. Thank you again for your valuable feedback!

---

### Official Review · Reviewer_cknv · 2024-10-28

**Soundness:** 3
**Presentation:** 3
**Contribution:** 3
**Rating:** 8
**Confidence:** 3

**Summary:**

The paper presents a novel approach to Preference Optimization for LLMs, named MallowsPO. The approach is inspired by the Mallow's theory of preferences ranking which assumes the preference functions to follow specific distributions parametrized by a dispersion index. Importantly, the dispersion index is prompt-dependent and models the fact that different prompts may have more or less concentrated preference distributions. The authors derive "Direct PO"-like versions of their approach and provide a way to estimate dispersion indices by Shannon entropy. Experiments show that MallowsPO outperform DPO.

**Strengths:**

I believe the paper has the following main strengths:
- The problem studied (i.e. LLM alignment to preference data) is very relevant
- The connection with Mallow's ranking models is, to the best of my knowledge, novel and meaningful
- The derived approach is sound
- Experiments are somewhat extensive and demonstrate the benefit of the introduced approach

**Weaknesses:**

Weaknesses are:
- The authors could do a better job at illustrating qualitative differences between policies trained with DPO and MallowsPO. In particular, one could plot entropy during training, number of tokens, distribution of rewards (when available) to make their case stronger and to understand what are the concrete differences between the two approaches. Part of this is left as future work, but I believe it would be nice to discuss/analyze it in the current paper
- The discussion and estimation of the dispersion index is not fully clear to me. Also, would have been nice to ablate the sensitivity of MallowsPO to other possible estimation approaches (or variants of chosen one).

**Questions:**

I do not fully understand the motivation behind the dispersion index and its estimate.
In particular, the index is estimated using the per-prompt entropy of the reference policy, but:
- on how many generation? does this require drawing more samples for an accurate estimate?

More importantly,
- Shouldn't the dispersion index indicate the dispersion of the unknown preference model (i.e. humans)? If so, I do not see why one should use the entropy of the SFT model (which we would still like to align to human preferences) to estimate such index.

---

> ### Author Response · Authors · 2024-11-19
>
> We sincerely appreciate your kind words on the novelty and soundness of our work. We also thank you for providing insightful questions, here we address them below:
>
> > The discussion and estimation of the dispersion index is not fully clear to me. ablate the sensitivity of MallowsPO to other possible estimation approaches
>
> The dispersion index $\phi(x)$ basically indicates how spread the completion distribution $P(y|x)$ is for a specific prompt $x$. To estimate $\phi(x) \in (0, 1]$, we consider using the entropy of $P(y|x)$. However, computing this exactly is difficult. Therefore, we come up with the following way to estimate it:
>
> $$ f(x) = \frac{1}{2}\cdot \frac{1}{\log (n)}  \cdot (H_{\text{win}}+ H_{\text{loss}})$$
>
> The intuition of this estimation comes from two parts:
>
> * **Entropy estimation.** Note that to the entropy of a generated sequence drawn from $P(Y|X)$  can be expressed as the sum of conditional entropy through the chain rule:
>  $$H (Y_1,Y_2,\cdots,Y_n\mid X) = H(Y_1|X) + H(Y_2|X, Y_1) + H(Y_3|X, Y_1, Y_2) + …..$$
> Therefore, we estimate above using the average of two realizations: winning and losing samples (which yields a non-biased estimator), which is
> $$H(\pi(\cdot\mid X))\approx \frac{1}{2} \sum_{i=1}^{N-1}\left[H(Y_{i+1}\mid Y_i=Y^w_i)+H(Y_{i+1}\mid Y_i=Y^l_i)\right].$$
>
> * **Upper bound.** In addition, because the dispersion $\phi(x) \in (0, 1]$, we normalize such a quantity by the upper bound of the entropy, which is $N\cdot log(|V|) = log(|V|^N) $, where $N$ is the max length and $|V|$ is the size of vocabulary (possible tokens), so that $f(x) \in (0,1]$. $|V|^N$ can also be understood as the total number of possible sequence of length N.
>
> We then estimate $-2\log(\phi(x))$ with $\phi\cdot \log(f(x))$. We empirically verified that such an estimator aligns well with the true $-2\log(\phi(x))$, shown in Figure 3, supposing we know the $\phi^*$. An immediate question is: How do we know determine $\phi^*$ in practice? Fortunately, a connection exists within the penalization of DPO. Specifically, while the penalty coefficient in DPO is represented by $\beta$ while the penalty coefficient in MallowsPO-$\theta$ is $-2log(\phi(x))\beta$. To ensure consistency, we impose $\phi$ such that
>
> $$E_{x\sim X}\left[\phi\cdot \log(f(x))\right]= 1.$$
>
> For other dispersion estimators, to the best of our knowledge, we don’t think there are other scalable and efficient estimating approaches for the dispersion index, comparing to our method which induces minimum extra computational burden. We agree that developing other dispersion estimation technique could possibly enhance the improvement brought by MallowsPO, and would like to leave the pursuit of efficient and more accurate dispersion estimation techniques in the future works.
>
> >  on how many generation? does this require drawing more samples for an accurate estimate?
>
> Our proposed method does not require generating answers or sampling multiple times. Please see the above explanation of our way to estimate the dispersion.
>
> >  the dispersion of the unknown preference model (i.e. humans)? If so, I do not see why one should use the entropy of the SFT model (which we would still like to align to human preferences) to estimate such index.
>
> Theoretically, the base model is typically trained on human-generated data such as news articles, textbooks or Wikipedia, which inherently reflect human preferences. Assuming that we may get a multimodal distribution for answers to a fixed prompt, SFT is known to guide the format of response which helps the language model determine the response format we desire and specify a subset of modals in human preferences. Therefore, even though the SFT model has not fully been aligned to given preferences, it should still, to some extent, reflect the broader spectrum of human preferences embedded within its pre-training data. Our proposed dispersion estimators based on this assumption indeed have shown the capability to capture the dispersion of the prompts empirically, as shown by Figure 1.

---

> > ### Comment · Reviewer_cknv · 2024-11-25
> >
> > I thank you the authors for their response, it fully answers my questions.
> > I have decided to raise my score.

---

### Official Review · Reviewer_Nu5X · 2024-11-03

**Soundness:** 2
**Presentation:** 2
**Contribution:** 2
**Rating:** 5
**Confidence:** 3

**Summary:**

This paper introduces a preference optimization method called MallowsPO, which extends DPO to incorporate human preference diversity. Authors leverage Mallows’ theory of preference ranking and propose a dispersion index that accounts for variations in human agreement, allowing diverse human preferences on prompt responses. Empirical studies are performed on benchmark tasks to show the performance by using the proposed dispersion index. Comparisons are done with other SOTA offline preference optimization methods.

**Strengths:**

This paper presents an extension to DPO with theoretical guarantees and demonstrates practical improvements in LLM fine-tuning on a set of benchmark tasks.  The motivation behind is clearly justified and the studied setting is of practical interests.

The introduction of preference dispersion through Mallows’ ranking theory is interesting in preference optimization and is presented clearly. The paper could, however, benefit from additional clarity in Section 3.1, where the transition from traditional DPO to MallowsPO is described. Specifically, more examples or visual aids could assist readers unfamiliar with Mallows’ theory.

The paper provides a detailed decomposition of the reward function based on dispersion, which is mathematically well-grounded and introduces two variations (MallowsPO-$\theta$ and MallowsPO-$\phi$) using different discrepancy functions. This addition allows for flexibility and generalization beyond existing DPO methods.

**Weaknesses:**

The use of a dispersion index adds interpretability to preference-based tuning, yet further details on how practitioners could leverage these insights in real-world applications could enhance the paper’s practical relevance.

While the empirical results are promising, additional insights into the MallowsPO model’s performance under varying preference dispersion scenarios (e.g., low vs. high dispersion prompts) could strengthen the evaluation. This might highlight more specific cases where MallowsPO demonstrates clear advantages over other DPO variants.

**Questions:**

See above

---

> ### Author Response · Authors · 2024-11-16
> **Improved manuscript with more discussions on Mallows Ranking Model and additional insights**
>
> We sincerely appreciate your kind words about the clarity and interest of our work. We are also grateful for your valuable comments, which will help us enhance the quality of our paper. Below, we address your questions and suggestions in detail:
>
> > additional clarity in Section 3.1
>
> Thank you for your suggestion on providing more background on Mallows Ranking Model. We have revised Section 3.1 to enhance the intuitive understanding of the Mallows model. Concretely, we add a 3-items illustrative example and compute their discrepancy function values under Mallows-$\theta$ and Mallows-$\phi$ ranking models separately in Section 3.1 (we mark the added paragraph in blue). We also include more details about this example, including computations, visual plots of probability density function, and their changes with respect to the dispersion parameter $\phi$ in the Appendix C. For transition from traditional DPO to MallowsPO under Mallows Ranking Model, please see the two paragraphs after Corollary 2.
>
> > further details on how practitioners could leverage these insights in real-world applications could enhance the paper’s practical relevance.
>
> For practical relevance, we consider the following directions on further leveraging these insights in real-world applications, focusing on *curriculum learning* and *personalized alignment* through contextual scaling. We outline these future directions in the conclusion section.
>
> **Curriculum learning**. Curriculum learning is a training strategy inspired by human learning processes, where models are trained on tasks or data organized from simpler to more complex examples, instead of presenting all examples randomly or simultaneously. This approach has been shown to guarantee faster training convergence [1], better generalization [2] and improved model robustness [3]. Within our framework, our proposed dispersion index can be utilized to indicate the determinism or diversity of the response to different questions, offering guidance on the complexity of the prompts to achieve curriculum learning.
>
> **Personalized Alignment**. Furthermore, by considering $\phi(x, p)$, where $p$ is a personal feature, our method can be adjusted to consider a personalized preference ranking model. Note that how to achieve this with DPO is not straightforward. However, it is more natural in MallowsPO because of the existence of dispersion parameter $\phi(x)$ in the Mallows model, which controls how spread out the distribution should be. In this context, dispersion can be extended beyond the prompt level to encompass different users or user groups $p$. In terms of the preference optimization objective, different contextual scaling on the penalization coefficient can be considered to accommodate varying user or user group preferences for responses generated by the Supervised Fine-Tuning (SFT) model. Contextual scaling, tailored to different user preferences, enables a customized user experience.
>
> Both curriculum learning and personalization through contextual scaling present promising directions for further development and can be expanded based on our foundational research. However, these ideas merit dedicated presentation and discussion in separate works and are thus reserved for future exploration. We have updated these discussions in **Section 6 (Conclusion)**, which are marked in blue, and in **Appendix F**.
>
> > additional insights into the MallowsPO model’s performance under varying preference dispersion scenarios (e.g., low vs. high dispersion prompts) could strengthen the evaluation
>
> The key insight behind MallowsPO's performance under varying preferences, which is a natural phenomenon in reality, is its explicit incorporation of an inductive bias that captures dispersion information for the preference model, which leads to improved evaluation performance. This approach contrasts with the original DPO, which does not account for this aspect. While it is possible to rediscover the Mallows model using the Bradley-Terry (BT) model by redefining a new reward function as $\tilde{r}(\cdot; x)=\phi(x)⋅r(\cdot; x)$, MallowsPO directly builds a structure that effectively channels dispersion information for improved modeling. Leveraging this structural insight of the preferences has already shown clear empirical gain for LLM preference optimization in our work. We would like to investigate theoretically provable benefits of MallowsPO over vanilla DPO without dispersion in future work.
>
> ---
>
> **Reference**.
>
> [1] Bengio, Y. et.al. (2009, June). Curriculum learning. In Proceedings of the 26th annual international conference on machine learning (pp. 41-48).
>
> [2] Graves, A. et.al. (2017, July). Automated curriculum learning for neural networks. In international conference on machine learning (pp. 1311-1320). PMLR.
>
> [3] Weinshall, D. et.al.. Curriculum learning by transfer learning: Theory and experiments with deep networks. In International conference on machine learning (pp. 5238-5246). PMLR.

---

> ### Author Response · Authors · 2024-11-25
>
> Dear Reviewer Nu5X,
>
> We hope this message finds you well. We kindly want to check if you had a chance to review our rebuttal, and if you have any further questions or comments we can address to help with your evaluation. Thanks again for your efforts and suggestions in improving our manuscript.
>
> Sincerely,
>
> The authors

---

> ### Author Response · Authors · 2024-11-27
> **Followup**
>
> Dear Reviewer Nu5X,
>
> We hope this message finds you well. As the deadline for revising the manuscript approaches on November 27th at 11:59 PM AoE, we are writing to follow up again to check if we have addressed your concerns to help with your evaluation. Your feedback has been invaluable in refining our work, and we sincerely appreciate the time and effort you’ve dedicated to improving our manuscript.
>
>
> Sincerely,
>
> The authors

---

> ### Author Response · Authors · 2024-11-27
> **Friendly Reminder: Revision Deadline is today**
>
> Dear Reviewer Nu5X,
>
> As today is exactly the revision deadline, we kindly request your feedback on our rebuttal. We would greatly appreciate it if you could let us know whether we have successfully addressed your concerns. Thank again for the time and effort you’ve dedicated to improving our manuscript.
>
> Sincerely,
>
> The authors

---

> ### Comment · Reviewer_Nu5X · 2024-11-27
>
> I thank the authors for their responses. After going through the responses and the revised draft again, here are some questions that I still have:
> 1. What is the function $g$ in Thm 1 (Line 203)? It was not introduced in previous context nor in the proof. Is it meant to be a link function? If it is, it is assumed to be a specific link function? What properties it should satisfy for Thm 1 to hold? It needs to be clearly articulate before introducing the extra notations.
>
> 2. I appreciate the authors' effort in providing examples in the main text and appendices. However, further insights (not necessarily to be long) should be provided. The added example highlighted in blue (Line 169 - 177) is purely calculation. Audience will benefit further if authors can add a sentence or two for the key takeaways.
>
>    - In particular, given the same central ranking and the same observation, different Mallows model lead to different dispersion. How do we compare these results?  Can we safely draw consistent conclusions with different Mallows model? e.g. If a completion $y_1$ is better than $y_2$ under one Mallows model, does it suggest $y_1$ is always better than $y_2$ under other Mallows model?
>    - From my understanding, any Mallows model cannot perfectly capture / model the human preference, BT model likewise. As a result, in reality, how do we choose which Mallows model to used for preference learning? Can authors comments under which situations do one model excel the other? Is there any guidline?
>    - Do Mallows models always excel BT model in better capturing human preference? Can we prove it theoretically (may not be easy)? Or authors can comment whether Mallows models are more general in the sense that pairwise preference data is not necessary as in BT model etc.
>
>     Such above insights will be more informative.
>
> 3.  I appreciate the authors' responses in discussing the possible extensions regarding the practical applicability. However, extending to curriculum learning and personalized alignment will be far beyond the scope of the current manuscript. What I wonder is that, in practice,
>       -  Whether the proposed method can be easily implemented as DPO? From authors' response to other reviewers, if MallowsPO is equivalent to DPO+disparity, does it suggest that only small modifications are expected?
>       - From Fig. 9, it seems MallowsPO can converge faster than DPO. Can authors comment whether this computational benefit can be always achieved? If yes, can we prove the faster convergence can be indeed achieved?
>       - Compared to DPO, does MallowsPO brings significantly extra computation overhead in order to compute the dispersion? Empirical comparisons in terms of computation will be benefitial.

---

> ### Author Response · Authors · 2024-11-28
>
> We sincerely thank you for further questions and comments to improve our manuscript. We have updated our manuscript per your suggestions, and below we also address or answer your concerns one by one.
>
> **Questions in Point 1 raised by reviewer**
>
> > What is the function $g$ in Thm 1 (Line 203)? It was not introduced in previous context nor in the proof. Is it meant to be a link function? If it is, it is assumed to be a specific link function?
>
> Yes, it is meant to be the link function, which implicitly appeared in the proof of Theorem 1. We have edited the descriptions of Theorem 1 (also the paragraph before it) to have a clearer explanation of the assumption and role of this link function.  We have also marked our changes in orange for your reference. Thanks for pointing this out!
>
>
> **Questions in Point 2 raised by reviewer**
>
> Kindly refer to the added intuition in Section 3.1 of the revised manuscript, as well as our responses to the reviewer’s questions:
>
> > Can we safely draw consistent conclusions with different Mallows model
>
> The answer is NO. Different discrepancy functions yield different Mallows Ranking Model, and we can easily find examples of inconsistency between different Mallows Ranking Models. Take a four item example, if the true ranking is $\mu_0=(1,2,3,4)$, consider two ranks $\mu_1= (2,1,4,3)$, $\mu_2=(1,4,2,3)$, then we can compute that for Spearman’s rho discrepancy (utilized in Mallows-$\theta$) $d(\mu_0,\mu_1)=4<d(\mu_0,\mu_2)=6$, however the Kendall’s tau (utilized in Mallows-$\phi$) can be computed as $d(\mu_0,\mu_1)=2=d(\mu_0,\mu_2)$. This showcases that different Mallows Ranking Models are not consistent because of their difference in discrepancy function. We have updated more details on this computation in Appendix C.4 for your reference. We have also included this conclusion in the introduction of Mallows Ranking Models near the end of Sc3.1, providing more insights per your suggestion.
>
> > As a result, in reality, how do we choose which Mallows model to use for preference learning? Can authors comment under which situations do one model excel the other? Is there any guideline?
>
> Thanks for raising this point. We have to admit that we haven’t found clear guidelines on choosing among Mallows Ranking Models, namely Mallows-$\theta$ and Mallows-$\phi$. For simplicity of implementation, we would recommend to always first try the Mallows-$\theta$ model, which is easy to modify upon codes of DPO and implement, and we have also demonstrated the effectiveness and clear advantages of MallowsPO over DPO baseline empirically. If one does not have a concern of computational budget, he/she can always try both MallowsPO-$\theta$ and MallowsPO-$\phi$ provided in our paper.
>
> > Do Mallows models always excel BT models in better capturing human preference? Can we prove it theoretically (may not be easy)? Or authors can comment whether Mallows models are more general in the sense that pairwise preference data is not necessary as in BT model etc.
>
> For this, if we focus on the application of aligning language models from human preferences and the perspective of designing loss objectives, we are pretty confident that the answer is yes. Essentially, MallowsPO incorporates an extra freedom, “dispersion” in the loss objective, which is the central concept of our paper and allows a more expressive characterization of human preferences. One can always choose a constant 1 as dispersion, which will then yield DPO since DPO does not consider dispersion. This shows that DPO is a special instance of MallowsPO-$\theta$, thus MallowsPO indeed can yield better performance, both theoretically and empirically.
>
> **Questions in Point 3 raised by reviewer**
>
> > Whether the proposed method can be easily implemented as DPO?
>
> Yes, the implementation is as easy as DPO and we have also already included the code in the supplementary materials. MallowsPO is pretty straightforward to implement, thanks to our choice of dispersion estimator. Long story short, our estimator needs to compute the entropy from the logits, which is a direct product in the computation of DPO objective, thus there is minimum extra computational burden.
>
> > Can authors comment whether this computational benefit can be always achieved?
>
> Thanks for observing this and pointing it out. Throughout this paper, we have been focusing on comparing the final policy DPO and MallowsPO converge to, and we indeed did not compare the rate of convergence to the policy.  We are thus not sure about your question since that’s not our paper’s primary claim or focus, but we would like to pursue this in future works.
>
> >  does MallowsPO bring significantly extra computation overhead in order to compute the dispersion?
>
> Similar to the question earlier, there is a minimum extra computational burden introduced by MallowsPO over the DPO.

---

> > ### Author Response · Authors · 2024-11-30
> >
> > Dear Reviewer Nu5X,
> >
> > Thank you for your time, effort, and valuable feedback. As the discussion period comes to an end, we would like to confirm if our responses from a few days ago have addressed your concerns.
> >
> > We are also more than happy to assist if you have further questions.
> >
> > Thank you again for your thoughtful engagement!
> >
> > Best regards,
> >
> > The Authors

---

> > > ### Author Response · Authors · 2024-12-02
> > >
> > > Dear Reviewer Nu5X,
> > >
> > > Thank you once again for your thoughtful feedback and engagement during the review process. We wanted to follow up again as today marks the final day of the author-reviewer discussion period.
> > >
> > > If you have any additional questions or require further clarification before the discussion closes, please feel free to let us know, and we will do our best to assist promptly.
> > >
> > > We greatly appreciate your time and effort in reviewing our work.
> > >
> > > Best regards,
> > >
> > > The Authors

---

> > > > ### Comment · Reviewer_Nu5X · 2024-12-03
> > > >
> > > > Thm 1 assumes the existence of a link function that satisfy the highlighted conditions. In reality, which link function achieves it? Do the linked functions need to satisfy (extra) "nice properties" such as convexity etc. in order to satisfy the assumption?

---

> ### Author Response · Authors · 2024-12-03
>
> We sincerely thank you for the further question and discussions.
>
> We have shown in Corollary 2 that when the discrepancy function $d(\cdot,\cdot)$ is chosen as Spearman’s rho (i.e. Mallows-$\theta$ model), the link function is $$g(y)=\sigma(-2\log\phi(x)\cdot y),$$
> which is a scaled sigmoid function also defined in Equation (8). In Corollary 3, when $d(\cdot,\cdot)$ is chosen as Kendall’s tau (i.e. Mallows-$\phi$ model), the link function $g(\cdot)$ is a bit more complicated, which is defined after Equation (9).
>
> Back to reviewer's general question about Thm1, as far as we know, there is not a simple closed-form mapping from any discrepancy function $d(\cdot,\cdot)$ to a link function in the Mallows setting. For some discrepancy functions $d(\cdot,\cdot)$, it's possible that corresponding link function that satisfies the condition we propose in Thm 1 may not exist. Fortunately, for Mallows-$\theta$ and Mallows-$\phi$, such link functions do exist and yield a closed form. Thus we do not required extra conditions on link functions like convexity.

---

> ### Author Response · Authors · 2024-12-03
>
> Out of the scope of the project, in general any increasing link function that takes value in (0,1) should work for practical optimization, which is satisfied by sigmoid (DPO), scaled sigmoid (Mallows-$\theta$) and the link function of Mallows-$\phi$.
>
> Though our Thm1 does not need extra properties of the link function, we also would like to mention that if we need the model to have additional "nice" properties, it may require log-concavity, see e.g. [Tang 2020](https://projecteuclid.org/journals/electronic-journal-of-statistics/volume-14/issue-2/The-existence-of-maximum-likelihood-estimate-in-high-dimensional-binary/10.1214/20-EJS1766.full) which discussed the MLE estimation of the binary response/preference models. This indeed follows your intuition that better properties of link functions like convexity could yield theoretical benefits (also possibly in practice for optimization). We will add this comment in a future version of the paper. Thanks for the enlightening question.

---

> ### Author Response · Authors · 2024-12-03
>
> Dear Reviewer Nu5X,
>
> We sincerely appreciate your efforts in improving our manuscript and active discussion with us. Since the discussion period will end within hours, we would sincerely appreciate it if you could kindly consider adjusting your scores if we have addressed your concerns. Thanks again!

---

> > ### Comment · Reviewer_Nu5X · 2024-12-03
> >
> > Thank you for your replies.
> >
> > I appreciate the authors’ active responses for rebuttal. The motivation and the setting of the paper is indeed interesting and of practical interest. Currently the manuscript provides a first-step analysis to apply Mallow’s theory in addressing the problem of preference dispersion. While the main idea of the paper can be useful, it will benefit the audience much further if the manuscript could provide deeper insights as outlined earlier. As such, I prefer maintaining my original assessment at this point, and will further discuss with other reviewers and AC.

---

> ### Author Response · Authors · 2024-12-03
>
> We sincerely thank you for your active engagement during the rebuttal period, and appreciate for your emphasizing that our paper is interesting and of practical interest to apply Mallow’s theory in addressing the problem of preference dispersion.
>
> Our work is the first to consider the ranking model beyond BT setting which already shows clear empirical gain, and we believe that can also entrigger a line of future research to further utilize the concept of dispersion (including directions like personal alignment and curriculum learning we mentioned earlier). We agree with the reviewer that deeper insights can benefit our paper. Given that our rebuttal answers are rather long (we try to be as detailed as our best), we want to summarize here (also highlight) as a friendly reminder that in our revision:
>
> * We have included the insights on **whether different Mallows model are consistent** (raised by the reviewer's first response) detailedly with a concrete example in **Appendix C.4**.
>
> * For insights on difference between BT and Mallows, we emphasized in our earlier response and discussed throughout our paper that **DPO can be treated as a special instance for MallowsPO-$\theta$** by treating dispersion index as a constant, which is the primary interest of this paper for LLM alignment/preference optimization. Thus **Mallows model (which yields MallowsPO) indeed `exceeds' BT (which yields DPO) as it effectively channels dispersion information to improve modeling**. As a remark, however, we tend not to make claims on Mallows exceeding BT from a pure ranking model perspective, as they represent fundamentally different approaches to modeling rankings, and their comparative merits have been debated for decades. Additionally, the complexity of human preferences further complicates such comparisons. Moreover, such a comparison lies outside the primary focus of our work. Nevertheless, we are willing to include more theoretically rigorous arguments and discussions of comparing ranking models that already appear in the literature in the camera ready version if accepted.
>
> We sincerely hope that our summarization of answers and revision could help for your final assessment, and thanks again for your efforts on improving our manuscript.

---

### Official Review · Reviewer_x8zh · 2024-11-05

**Soundness:** 3
**Presentation:** 3
**Contribution:** 3
**Rating:** 8
**Confidence:** 3

**Summary:**

This work introduces a novel technique for fine-tuning LLMs called MallowsPO, which is inspired by Mallows' theory of preference ranking. MallowsPO utilizes a dispersion index to reflect the dispersion of human preference to prompts. Existing DPO models are special cases of this dispersion index. Additionally, MallowsPO is compatible with other SOTA offline preference optimization methods.

**Strengths:**

1. This paper is clearly written and well structured. The authors provide sufficient analytical and empirical comparisons with previous methods.

2. The contribution to the methodology for fine-tuning large language models is significant. This work proposes a novel preference optimization framework based on Mallows' preference ranking theory beyond the BT setting, which provides a novel perspective on preference modeling. This new method introduces an important component, the dispersion index, to address the shortcomings of DPO in characterizing the diversity of human preferences. This work is technically sound and offers a detailed theoretical analysis. MallowsPO is a generalization of the well-known DPO method and can cover DPO as a special case under a certain parameter setting.

3. Besides its theoretical contribution, this work also has significant empirical contributions. It proposes empirical approximations to the dispersion index to facilitate computation. Moreover, extensive experiments for the proposed methods are conducted under various setups, including synthetic bandits, controllable generation, fine-tuning Pythia 2.8B on off-policy Anthropic HH dataset, and fine-tuning Llama3-8B-Instruct on a hybrid UltraFeedback dataset, to show the advantages of MallowsPO. Moreover,  MallowsPO is a sufficiently general approach that can be integrated with other offline preference optimization methods, such as simPO and IPO, which have better performance compared with their vanilla counterparts.

**Weaknesses:**

None.

**Questions:**

None

**Details Of Ethics Concerns:**

None.

---

> ### Author Response · Authors · 2024-11-16
>
> We sincerely thank you for your thoughtful feedback. We are glad you found our work clear, well-structured, and a meaningful contribution to LLM fine-tuning. Your recognition of the MallowsPO framework, its theoretical and empirical advancements is greatly appreciated.

---

### Meta-Review · Area_Chair_1dwX · 2024-12-18

**Metareview:**

This paper proposes a novel approach to preferential optimization based on Mallows' theory of preference ranking. One benefit over DPO is that the dispersion index in the model can be prompt specific and characterize the hardness of ranking for that prompt. This paper is well written, well motivated, the method is grounded in theory, and it is comprehensively evaluated. The topic is timely. The scores of this paper are 2x 8 and 2x 5, which is an improvement over the initial 8, 6, 5, and 3. The discussion clarified several aspects of the method, such as differences from DPO and the loss of reward difference information. Please incorporate these in the camera-ready paper and discuss shortcomings of your method. This paper is accepted.

**Additional Comments On Reviewer Discussion:**

See the meta-review for details.

---

### Decision · Program_Chairs · 2025-01-22

Accept (Poster)